# Invariant Causal Set Covering Machine

**Thibaud Godon**
*thibaud.godon.1@ulaval.ca*
*Université Laval*
**Baptiste Bauvin**
*Université Laval*
**Pascal Germain**
*Université Laval*
**Jacques Corbeil**
*Université Laval*
**Alexandre Drouin**
*ServiceNow Research*

**Reviewed on OpenReview:** *https: // openreview. net/ forum? id= slquR2A8rA*

## Abstract

Rule-based models, such as decision trees, appeal to practitioners due to their interpretable nature. However, the learning algorithms that produce such models are often vulnerable to spurious associations, and thus, they are not guaranteed to extract causally relevant insights. This limitation reduces their utility in gaining mechanistic insights into a phenomenon of interest. In this work, we build on ideas from the invariant causal prediction literature to propose *Invariant Causal Set Covering Machines*, an extension of the classical Set Covering Machine (SCM) algorithm for conjunctions/disjunctions of binary-valued rules that provably avoids spurious associations. The proposed method leverages structural assumptions about the functional form of such models, enabling an algorithm that identifies the causal parents of a variable of interest in polynomial time. We demonstrate the validity and efficiency of our approach through a simulation study and highlight its favorable performance compared to SCM in uncovering causal variables across real-world datasets.

## 1 Introduction

In some fields of application of machine learning, the use of learned models goes well beyond prediction. Domain experts often rely on a deeper inspection of such models to extract mechanistic insights into complex systems. For instance, in healthcare, one can train a model to predict predisposition to a disease based on genomics data (Szymczak et al., 2009). Significant insights can then be obtained by understanding which genomic traits are used for prediction (biomarkers). These constitute potential causes of the disease, which might be relevant targets in the elaboration of new therapies or drugs.

One kind of machine learning model that has been shown to allow for such scrutiny is rule-based models, such as decision trees (Breiman et al., 1984). Such models make inferences by applying a series of binary-valued rules to their input (e.g., the presence or absence of a mutation). In addition to their high level of interpretability, such models have been shown to scale particularly well to large feature sets and resist overfitting in the small data regime that is common in applications of machine learning to healthcare (Drouin et al., 2019). However, one must not be fooled by the ease of interpretation of such models, since the variables used for prediction may very well be spuriously associated with the outcome of interest.

In this work, we make a step towards alleviating this issue by proposing *Invariant Causal Set Covering Machines*, an extension of the Set Covering Machine algorithm (Marchand & Shawe-Taylor, 2002) for conjunctive and disjunctive classifiers, that avoids, as much as possible, relying on spurious associations for

prediction. Our work builds on previous advances in invariant causal prediction (Peters et al., 2016; Heinze-Deml et al., 2018; Bühlmann, 2020), where data is assumed to be collected in multiple environments (e.g., populations from different geographic locations, measurement devices with different calibrations, etc.).

**Contributions:**

1. We propose the Invariant Causal Set Covering Machines (ICSCM), an extension of the Set Covering Machine algorithm (Marchand & Shawe-Taylor, 2002) that relies on invariant causal prediction to avoid spurious associations (Section 3).

2. We support this new algorithm with theoretical results expressing conditions under which the causal parents of an outcome of interest can be recovered in polynomial time (Theorem 3.1).

3. We conduct an empirical study with simulated data to verify the correctness of the theory and the efficiency of the algorithm (Section 4.1), and then show, using real data, that the ICSCM tends to identify more causal parents than its SCM counterpart (Section 4.2).

## 2    Background and Related Work

### 2.1    Problem setting

We consider a standard supervised binary classification setting, where the observations are feature-label pairs $(\mathbf{x}, y)$, with $\mathbf{x} \in \mathcal{X}$ and $y \in \{0, 1\}$. Further, as in Heinze-Deml et al. (2018), we consider the case where the observations have been collected in multiple environments $e \in \mathcal{E}$, which correspond to various experimental conditions (e.g., populations from different geographic locations, measurement devices with different calibrations, etc.). Hence, we assume access to observations $(\mathbf{x}, y, e) \sim P(\mathbf{X}, Y, E)$, where the data-generating process $P$ factorizes according to $G$ depicted at Fig. 1. The random variable $\mathbf{X}$ is segmented as $\mathbf{X} = [\mathbf{X}_A, \mathbf{X}_B, \mathbf{X}_C]$, respectively denoting the causal parents of $Y$, variables that are not directly related to $Y$, and the causal children of $Y$. Formally, we make the following assumptions, which are common in the causal discovery literature (see Glymour et al., 2019):

**Assumption 2.1.** (Causal Markov assumption) We assume that *d-separation*[1] in $G$ implies conditional independence in $P$, i.e., for arbitrary random variables $U, V, W$:

$$U \perp\!\!\!\perp_G V \mid W \;\Rightarrow\; U \perp\!\!\!\perp_P V \mid W\,,$$

where $\perp\!\!\!\perp_G$ denotes d-separation and $\perp\!\!\!\perp_P$ denotes independence in distribution.

**Assumption 2.2.** (Faithfulness assumption) For arbitrary random variables $U, V, W$:

$$U \perp\!\!\!\perp_P V \mid W \;\Rightarrow\; U \perp\!\!\!\perp_G V \mid W\,.$$

Further, we assume that the environments are defined as follows, matching the setting of Heinze-Deml et al. (2018).

**Assumption 2.3.** (Environments) The environment $E$ is a causal parent of $\mathbf{X}$, but it is not a causal parent of $Y$ (see Fig. 1). In other words, the structural equation $Y := f(\mathbf{X}_A, \epsilon)$, with noise $\epsilon \perp\!\!\!\perp_P \mathbf{X}_A$, is invariant across environments.

Intuitively, this means that the distribution of features $\mathbf{X}$ can change across environments, but the mechanism that produces $Y$ from its causal parents $\mathbf{X}_A$ must remain stable.

**Goal:**    We aim to learn a classifier $h : \mathcal{X} \to \{0, 1\}$, such that $h(\mathbf{x}) = h(\mathbf{x}_A) = y$ with high probability, i.e., that closely approximates $Y$ while relying solely on its *causal parents* $\mathbf{X}_A$ and no other *spurious association*.

Moreover, we are interested in learning classifiers that are conjunctive in nature, so we make the following additional assumption on the functional form of $Y := f(\mathbf{X}_A, \epsilon)$:

---

[1]See Koller & Friedman (2009) (Chap. 3) for an introduction.

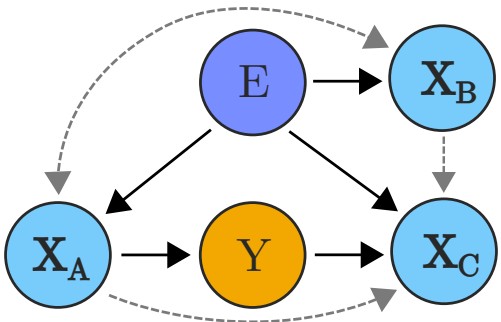

Figure 1: Graphical assumptions: the edge between $\mathbf{X}_A$ and $\mathbf{X}_B$ can be oriented in either way, but the resulting $G$ must be a Directed Acyclic Graph (DAG). Dashed edges are optional.

**Assumption 2.4.** (Functional form) The function $f$ is s.t.

$$f(\mathbf{X}_A, \epsilon) \stackrel{\text{def}}{=} g\big(r_1(\mathbf{X}_A, \epsilon_1) \wedge \ldots \wedge r_d(\mathbf{X}_A, \epsilon_d), \epsilon_g\big), \tag{1}$$

where $\langle r_1, \ldots, r_d \rangle$ are arbitrary binary-valued rules (e.g., threshold functions on the value of features), $\epsilon := \langle \epsilon_1, \ldots, \epsilon_d, \epsilon_g \rangle$ are noise terms sampled from arbitrary independent distributions, and $g$ is a function that tampers with the outcome based on $\epsilon_g$.

### 2.2 Set Covering Machines

We now introduce the Set Covering Machine (SCM) algorithm[2] (Marchand & Shawe-Taylor, 2002), which can be used to learn classifiers that are conjunctive in nature. Later on, at Section 3, we will explain how this algorithm can be extended to achieve the goal defined at Section 2.1.

Let $(\mathbf{x}, y)$ be a pair of features and binary label, as described at Section 2.1. The SCM algorithm is a greedy learning algorithm that attempts to find the shortest conjunction or disjunction,

$$h_{\text{conj}}(\mathbf{x}) = r_1(\mathbf{x}) \wedge \ldots \wedge r_d(\mathbf{x})$$
$$\text{or} \quad h_{\text{disj}}(\mathbf{x}) = r_1(\mathbf{x}) \vee \ldots \vee r_d(\mathbf{x}),$$

where the $r_i$ are binary-valued rules, that minimizes the expected prediction error:

$$\text{Err}_P(h) = \mathop{\mathbb{E}}_{(\mathbf{x},y) \sim P(\mathbf{X}, Y)} I[h(\mathbf{x}) \neq y],$$

where $I[\text{True}] = 1$ and 0 otherwise.

For conciseness, the rest of the presentation will focus on learning conjunctions ($h := h_{\text{conj}}$). This is not restrictive since any algorithm for learning conjunctions can also be applied to learning disjunctions by the simple application of De Morgan's law, i.e., $\neg\big(r_1(\mathbf{x}) \wedge r_2(\mathbf{x})\big) = \neg r_1(\mathbf{x}) \vee \neg r_2(\mathbf{x})$. Hence, all the forthcoming findings and observations equally apply to the case of learning disjunctions.

Algorithm 1 shows the pseudocode for learning a conjunction with the SCM algorithm. The algorithm starts with a data sample $\mathcal{S} = \{(\mathbf{x}_i, y_i)\}_{i=1}^m \sim P(\mathbf{X}, Y)^m$ and a set $\mathcal{R}$ of candidate binary-valued rules. It builds a conjunction by iteratively adding rules $r_i \in \mathcal{R}$. At each iteration, the best rule is selected based on a utility score $U_i$, which is a tradeoff between the number of negative examples correctly classified (i.e., covered by the rule[3]) and the number of positive examples misclassified by the rule. Then, the examples for which the model's outcome is settled (i.e., $h(\mathbf{x}) = 0$) are discarded, and the next iteration is performed considering the

---

[2]Not to be confused with **Structural Causal Models**. This low-probability clash is unfortunately beyond our control.
[3]Hence the name: Set *Covering* Machines

---

**Algorithm 1** Set Covering Machine

    **Input:** $\mathcal{S} = \{(\mathbf{x}_i, y_i, e_i)\}_{i=1}^m$, a data sample
    **Input:** $\mathcal{R}$, a set of candidate binary-valued rules
    **Input:** $p \in \mathbb{R}^+$, utility score hyperparameter
    **Input:** $n \in \mathbb{Z}$, conjunction length hyperparameter
    $\mathcal{P} \leftarrow \{(\mathbf{x}, y) \in \mathcal{S} \mid y = 1\}$          ▷ Positive examples
    $\mathcal{N} \leftarrow \{(\mathbf{x}, y) \in \mathcal{S} \mid y = 0\}$          ▷ Negative examples
    $\mathcal{H} \leftarrow \emptyset$          ▷ Rules in the conjunction
    **while** $|\mathcal{H}| < n$ and $\mathcal{N} \neq \emptyset$ **do**
        **for** each rule $r_i \in \mathcal{R}$ **do**
            $\mathcal{A}_i \leftarrow \{(\mathbf{x}, y) \in \mathcal{N} \mid r_i(\mathbf{x}) = y\}$          ▷ True negatives
            $\mathcal{B}_i \leftarrow \{(\mathbf{x}, y) \in \mathcal{P} \mid r_i(\mathbf{x}) \neq y\}$          ▷ False negatives
            $U_i \leftarrow |\mathcal{A}_i| - p \cdot |\mathcal{B}_i|$          ▷ Utility computation
        **end for**
        $i^\star \leftarrow \operatorname{argmax}_{1 \leq i \leq |\mathcal{R}|} U_i$
        $\mathcal{H} \leftarrow \mathcal{H} \cup \{r_{i^\star}\}$          ▷ Add best utility rule to the model
        $\mathcal{N} \leftarrow \mathcal{N} \setminus \mathcal{A}_{i^\star}$          ▷ Remove final samples
        $\mathcal{P} \leftarrow \mathcal{P} \setminus \mathcal{B}_{i^\star}$
    **end while**
    **Output:** The conjunction $h(x) = \bigwedge_{r \in \mathcal{H}} r(x)$

---

examples that remain to be classified.[4] The training stops when no negative examples remain to cover or when the maximum conjunction length $n$, which is a hyperparameter, is reached. Overall, the running time complexity of this algorithm is $O(m \cdot |\mathcal{R}| \cdot n)$.

The SCM algorithm is thus a simple and efficient way of learning conjunctive classifiers that minimize the expected prediction error. The learned predictive model has the benefit of being highly interpretable since its decision function is simple and typically relies on a few rules that can be inspected by domain experts (e.g., as in Drouin et al., 2019). However, it has one significant pitfall: nothing prevents $h(\mathbf{x})$ from relying on spurious associations between $\mathbf{X}$ and $Y$. In Section 3, we set out to alleviate this issue.

### 2.3 Invariant Causal Prediction

The two most related works from the causal inference literature are the seminal works of Peters et al. (2016) and Heinze-Deml et al. (2018) on invariant causal prediction. Both of these works rely on a multi-environment setting like the one described at Section 2.1. Moreover, both are based on the idea that the conditional distribution of $Y$, given all of its causal parents $\mathbf{X}_A$, should be invariant across environments; an idea that we also exploit in Section 3. In contrast with our work, Peters et al. (2016) require the structural equation between $Y$ and $\mathbf{X}_A$ to be linear, while we assume it to be conjunctive (see Theorem 2.4). As for Heinze-Deml et al. (2018), they consider non-linear structural equations, which are compatible with our setting. However, identifying $\mathbf{X}_A$ using their approach requires to perform conditional independence tests for all sets of potential parents, which requires $O(2^{|\mathbf{X}|})$ time, where $|\mathbf{X}|$ is the number of feature variables (see Section 2.1). In sharp contrast, we show that, by exploiting the conjunctive nature of the structural equation of $Y$, the causal parents $\mathbf{X}_A$ can be identified in polynomial time.

## 3 Invariant Causal Set Covering Machines

We now propose an extension of the classical Set Covering Machine algorithm that exploits invariances across environments (see Theorem 2.3) to learn classifiers that rely solely on the causal parents $\mathbf{X}_A$ of $Y$.

---

[4]Since $h$ is a conjunction, $h(\mathbf{x}) = 0$ if any $r_i(\mathbf{x}) = 0$. The outcome of the model cannot be changed by subsequent rules.

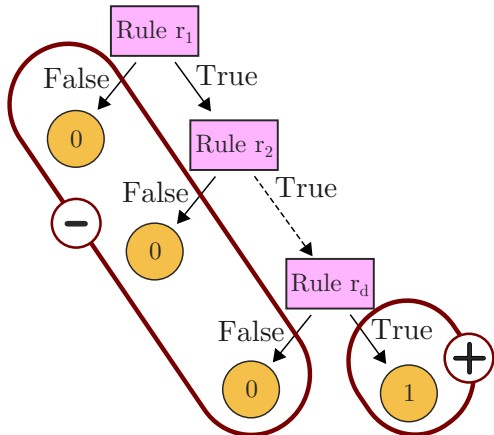

Figure 2: Tree-based representation of a $d$-rule conjunction. Positive and negative leaves are emphasized with $+$ and $-$, respectively. Notice that there are $d$ negative leaves and a single positive one, which is reached only when all rules are satisfied.

### 3.1 Tree-based representation

To facilitate the presentation, let us introduce an alternative perspective on conjunctions.

Let $f(x) \stackrel{\text{def}}{=} r_1(x) \wedge \ldots \wedge r_d(x)$ be an arbitrary conjunction of binary-valued rules $r_i$ applied to some input $x$, where each rule $r_i$ is a function of a single variable of $x$. If one assumes an ordering in the evaluation of the rules $r_i(x)$, then $f(x)$ corresponds to a simple decision tree composed of a single branch. This is illustrated at Fig. 2. Such a tree has $d$ *negative leaves* where $f(x) = 0$, which are attained when a rule $r_i(x) = 0$, and a single *positive leaf* where $f(x) = 1$, which is attained when $r_i(x) = 1 \; \forall i \in \{1, \ldots, d\}$.

With this in mind, we propose the following result:

**Theorem 3.1.** *(Model construction criteria) Assume that the data-generating process follows the causal graph depicted at Fig. 1 and that Assumptions 2.1, 2.2, 2.3, and 2.4 hold. Let*

$$f(\mathbf{X}^\star, \epsilon) = g(r_1(\mathbf{X}^\star, \epsilon_1) \wedge \ldots \wedge r_d(\mathbf{X}^\star, \epsilon_d), \epsilon_g) \,,$$

*with $\mathbf{X}^\star \subseteq \mathbf{X}$, be an arbitrary conjunction of $d$ binary-valued rules.*

*Without loss of generality, assume an arbitrary ordering of the rules $1 \ldots d$ and consider the tree-based representation depicted in Fig. 2. We have that, if:*

*(i) the distribution of $Y$ in the $i$-th negative leaf satisfies*

$$Y \perp\!\!\!\perp_P E \mid \mathbf{r}_{<i}(\mathbf{X}^\star, \epsilon_{<i}) = \mathbf{1}, r_i(\mathbf{X}^\star, \epsilon_i) = 0 \,, \tag{2}$$

*where $\mathbf{r}_{<i}(\mathbf{X}^\star, \epsilon_{<i}) = \mathbf{1}$ denotes that all rules preceding $r_i$ in the ordering have value 1, and*

*(ii) the distribution of $Y$ in the positive leaf satisfies*

$$Y \perp\!\!\!\perp_P E \mid \mathbf{r}_{<d}(\mathbf{X}^\star, \epsilon_{<d}) = \mathbf{1}, r_d(\mathbf{X}^\star, \epsilon_d) = 1 \,, \tag{3}$$

*then $\mathbf{X}_A \subseteq \mathbf{X}^\star$ and $\mathbf{X}_C \cap \mathbf{X}^\star = \emptyset$ .*

The proof below makes use of Theorem 2.1 and the conjunctive nature of $f$; if one of the rules has value $r_i(\mathbf{X}^\star, \epsilon_i) = 0$, then the values of the subsequent rules in the ordering have no impact on the final outcome of the conjunction.

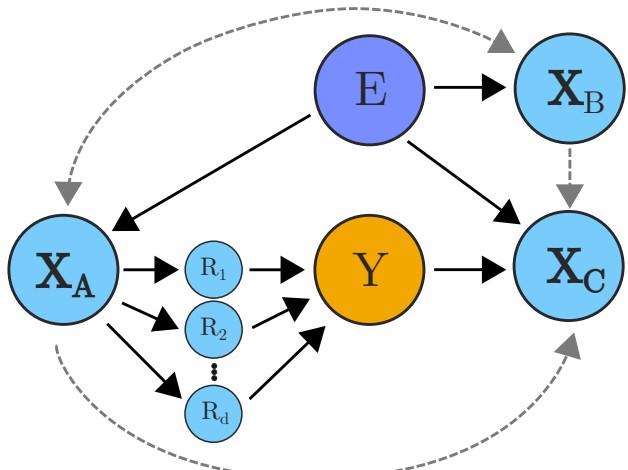

Figure 3: Implicit variables for binary-valued rules introduced in the proof of Theorem 3.1: this figure is an expanded version of the causal graph illustrated in Fig. 1 where the rules in the conjunction (Eq. 1) are represented as random variables that mediate all paths from $\mathbf{X}_A$ to $Y$.

*Proof.* The proof is divided in two cases and we proceed by contradiction. For simplicity, we use $R_i$ to denote the random variable $r_i(\mathbf{X}^\star, \epsilon_i)$. This amounts to introducing implicit variables for each binary-valued rule in the causal graph, as depicted by Fig. 3. In the following, we slightly abuse the notation by using $r_i$ to denote a value taken on by $R_i$.

Let us prove that properties *(i)* and *(ii)* implies $\mathbf{X}_A \subseteq \mathbf{X}^\star$ and $\mathbf{X}_C \cap \mathbf{X}^\star = \emptyset$ by contradiction. We consider two cases.

**Case 1: Suppose that the properties at Eq. (2) and Eq. (3) hold, but that $\mathbf{X}_A \nsubseteq \mathbf{X}^\star$.** We have that $\mathbf{X}_A \nsubseteq \mathbf{X}^\star$ and thus some of the causal parents of $Y$ are not in $\mathbf{X}^\star$, i.e., there exists $X_{A_i} \in \mathbf{X}_A$ such that $X_{A_i} \notin \mathbf{X}^\star$. Hence, we have that $Y \not\perp\!\!\!\perp_P E \mid \mathbf{X}^\star$, because there exists an unblocked path $E \to X_{A_i} \to Y$. Because $\mathbf{X}^\star$ does not contain all the causal parents of $R_1, \ldots, R_d$, we also have that $Y \not\perp\!\!\!\perp_P E \mid R_1, \ldots, R_d$. By the definition of conditional dependence, this means

$$\exists y, e, r_1, \ldots, r_d \text{ such that} \tag{4}$$
$$P(Y = y \mid R_1 = r_1, \ldots, R_d = r_d, E = e) \neq P(Y = y \mid R_1 = r_1, \ldots, R_d = r_d).$$

Recall that, given the conjunctive nature of $f(\mathbf{X}^\star, \epsilon)$, each combination of $r_1, \ldots, r_d$ corresponds to a leaf in the conjunction (see Fig. 2). Therefore, depending on the value of $r_1, \ldots, r_d$, we will reach a contradiction for either a negative or a positive leaf:

- **Negative leaves:** $\exists j, r_j = 0$, then we have

$$P(Y \mid R_1 = 1, \ldots, R_{j-1} = 1, R_j = 0, E = e) \neq P(Y \mid R_1 = 1, \ldots, R_{j-1} = 1, R_j = 0),$$

  which is in contradiction with Eq. (2).

- **Positive leaf:** $\forall j, r_j = 1$, then we have

$$P(Y = y \mid R_1 = 1, \ldots, R_d = 1, E = e) \neq P(Y = y \mid R_1 = 1, \ldots, R_d = 1),$$

  which is in contradiction with Eq. (3).

**Case 2 : Suppose that the properties at Eq. (2) and Eq. (3) hold, but that $\mathbf{X}_C \cap \mathbf{X}^\star \neq \emptyset$.** There are some descendants of $Y$ in $\mathbf{X}^\star$, i.e., there exists $X_{C_i} \in \mathbf{X}_C$ such that $X_{C_i} \in \mathbf{X}^\star$. Recall, from our

graphical assumptions (see Fig. 1), the existence of the v-structure $Y \to X_C \leftarrow E$. Since, $X_{C_i} \in \mathbf{X}^\star$, there exists at least one $R_j$ such that $X_{C_i} \to R_j$. Because conditioning on $R_j$ opens the path $E - X_{C_i} - Y$, we have that $Y \not\perp\!\!\!\perp_P E \mid R_1, \ldots, R_d$. By the definition of conditional dependence, Eq. (4) holds. Thus, by an identical argument to case 1, we reach a contradiction to Eq. (2) and Eq. (3) in negative and positive leaves, respectively.

□

## 3.2 Model construction

Theorem 3.1 gives us criteria that can be evaluated at each step of building a conjunction and that guarantee reliance on all $\mathbf{X}_A$ but none of the $\mathbf{X}_C$. We thus propose to modify the original SCM algorithm to

(i) prevent adding rules to the conjunction that would contradict the property of Eq. (2); and

(ii) stop adding rules to the conjunction when the property of Eq. (3) is satisfied.

The above criteria (i) and (ii) can be evaluated by statistical independence tests, up to some significance level $\alpha \in \mathbb{R}^+$, which is a hyperparameter of our proposed algorithm. Note that, in our experiments, we use a chi-squared ($\chi^2$) hypothesis test with $\alpha = 0.05$.

**Applying Criterion (i).** Consider the inner loop of Algorithm 1: Let $k = |\mathcal{H}|$ be the number of rules previously added to the conjunction, $r_i \in \mathcal{R}$ be a candidate rule, and $(\mathbf{x}_j^{(i)}, y_j^{(i)}, e_j^{(i)}) \in \mathcal{A}_i \cup \mathcal{B}_i$ denote the samples that would belong to a newly added negative leaf if $r_i$ were added to the conjunction. Denoting with $k' = k + 1$, we have

$$(y_j^{(i)}, e_j^{(i)}) \sim Y \times E \mid \mathbf{r}_{<k'}(\mathbf{X}^\star, \epsilon_{<k'}) = \mathbf{1}, r_{k'}(\mathbf{X}^\star, \epsilon_{k'}) = 0.$$

To implement Criterion (i), we simply disregard the rule $r_i$ if the p-value of a statistical independence test between the observed $y_j^{(i)}$ and $e_j^{(i)}$ from $\mathcal{A}_i \cup \mathcal{B}_i$ is lower than the threshold $\alpha$. Among the remaining rules, the one with the maximum utility score $U_i$ is added to the model.

**Applying Criterion (ii).** Consider the end of the outer loop of Algorithm 1. Let $k' = |\mathcal{H}|$ be the number of rules after the best rule $r_{i*}$ is added to the model. Then, the samples $(\mathbf{x}_j', y_j', e_j') \in \mathcal{P} \cup \mathcal{N}$ belong to the newly created positive leaf. We have

$$(y_j', e_j') \sim Y \times E \mid \mathbf{r}_{<k'}(\mathbf{X}^\star, \epsilon_{<k'}) = \mathbf{1}, r_{k'}(\mathbf{X}^\star, \epsilon_{k'}) = 1.$$

To implement Criterion (ii), we stop adding rules to the model (i.e., we exit the *while* loop) when the p-value of a statistical intendance test between the observed $y_j'$ and $e_j'$ from $\mathcal{P} \cup \mathcal{N}$ is greater than the threshold $\alpha$.

## 3.3 Model pruning

Note that Theorem 3.1 does not guarantee the *minimality* of $\mathbf{X}^\star$, i.e., Equations (2) and (3) could be satisfied even if $\mathbf{X}_B \cap \mathbf{X}^\star \neq \emptyset$. Hence, we propose the following pruning procedure, which can be applied as long as none of the rules $r_i$ jointly relies on elements from both $\mathbf{X}_A$ and $\mathbf{X}_B$. For example, this holds in the common setting where the rules are threshold functions on the value of a single feature.

**Proposition 3.2.** *(Pruning) Assume that Assumptions 2.1 and 2.2 hold. Let $f(\mathbf{X}^\star, \epsilon)$ be a conjunction that satisfies Eq. (2) and Eq. (3), but is non-minimal, i.e., $\mathbf{X}_B \cap \mathbf{X}^\star \neq \emptyset$. For any $\mathbf{X}^\dagger \in \mathbf{X}^\star$, we have that*

$$Y \perp\!\!\!\perp_P E \mid \mathbf{X}^\star \setminus \mathbf{X}^\dagger \quad \Longleftrightarrow \quad \mathbf{X}^\dagger \notin \mathbf{X}_A.$$

*Proof.* Since $f(\mathbf{X}^\star, \epsilon)$ satisfies Equations (2) and (3), we know that $\mathbf{X}_A \subseteq \mathbf{X}^\star$ and $\mathbf{X}_C \cap \mathbf{X}^\star = \emptyset$. We now consider two cases.

---

**Algorithm 2** Invariant Causal Set Covering Machine

---

**Input:** $\mathcal{S} = \{(\mathbf{x}_i, y_i, e_i)\}_{i=1}^m$, a data sample
**Input:** $\mathcal{R}$, a set of candidate binary-valued rules
**Input:** $p \in \mathbb{R}^+$, utility score hyperparameter
**Input:** $n \in \mathbb{Z}$, conjunction length hyperparameter
**Input:** $\alpha \in \mathbb{R}^+$, independence threshold hyperparameter

$\mathcal{P} \leftarrow \{(\mathbf{x}, y, e) \in \mathcal{S} \mid y = 1\}$ ▷ Positive examples
$\mathcal{N} \leftarrow \{(\mathbf{x}, y, e) \in \mathcal{S} \mid y = 0\}$ ▷ Negative examples
$\mathcal{H} \leftarrow \emptyset$ ▷ Rules in the conjunction
$\gamma \leftarrow 1$ ▷ A stopping criterion indicator

**while** $|\mathcal{H}| < n$ and $\mathcal{N} \neq \emptyset$ and $\gamma < \alpha$ **do**

    **for** each rule $r_i \in \mathcal{R}$ **do**
        $\mathcal{A}_i \leftarrow \{(\mathbf{x}, y, e) \in \mathcal{N} \mid r_i(\mathbf{x}) = y\}$ ▷ True negatives
        $\mathcal{B}_i \leftarrow \{(\mathbf{x}, y, e) \in \mathcal{P} \mid r_i(\mathbf{x}) \neq y\}$ ▷ False negatives
        $\pi \leftarrow$ p-value of a **independence test** between $y_j^{(i)}$ and $e_j^{(i)}$ from $\mathcal{A}_i \cup \mathcal{B}_i$. ▷ Section 3.2, Criterion (i)
        $U_i \leftarrow (|\mathcal{A}_k| - p \cdot |\mathcal{B}_k|)$ **if** $\pi > \alpha$ **else** $-\infty$
    **end for**

    $i^\star \leftarrow \operatorname{argmax}_{1 \leq i \leq |\mathcal{R}|} U_i$
    **if** $U_{i^\star} = -\infty$ **then** break ▷ Stop adding rules
    $\mathcal{H} \leftarrow \mathcal{H} \cup \{r_{i^\star}\}$ ▷ Add best utility rule to the model
    $\mathcal{N} \leftarrow \mathcal{N} \setminus \mathcal{A}_{i^\star}$ ▷ Remove final samples
    $\mathcal{P} \leftarrow \mathcal{P} \setminus \mathcal{B}_{i^\star}$
    $\gamma \leftarrow$ p-value of a **statistical test** between $y_j'$ and $e_j'$ from $\mathcal{P} \cup \mathcal{N}$. ▷ Section 3.2, Criterion (ii)
**end while**
[Optionally] Conduct **pruning** according to the procedure described at Section 3.3.
**Output:** the conjunction $h(x) = \bigwedge_{r \in \mathcal{H}} r(x)$

---

First, let us reason about sufficiency and show that

$$Y \perp\!\!\!\perp_P E \mid \mathbf{X}^\star \setminus \mathbf{X}^\dagger \;\Rightarrow\; \mathbf{X}^\dagger \notin \mathbf{X}_A\,.$$

This follows directly from the faithfulness assumption (Theorem 2.2), since taking $\mathbf{X}^\dagger$ from $\mathbf{X}_A$ would open at least one path from $E$ to $Y$ in $G$ (see Fig. 1), contradicting the conditional independence statement.

Second, let us reason about necessity and show that

$$\mathbf{X}^\dagger \notin \mathbf{X}_A \;\Rightarrow\; Y \perp\!\!\!\perp_P E \mid \mathbf{X}^\star \setminus \mathbf{X}^\dagger\,.$$

This follows directly from the causal Markov assumption (Theorem 2.1) since removing an element of $\mathbf{X}_B$ leaves all paths in $G$ from $E$ to $Y$ blocked. $\qquad\square$

Hence, the $f(\mathbf{X}^\star, \epsilon)$ can be pruned, iteratively, by applying a conditional independence test to each $\mathbf{X}^\dagger \in \mathbf{X}^\star$ and removing any rule that makes use of non-causal-parents of $Y$. For performing the pruning in the experiments of Section 4.1, we used the conditional G-test with significance level $\alpha = 0.05$.

### 3.4 The ICSCM algorithm

By combining the construction criteria of Section 3.2 (derived from Theorem 3.1) with the optional pruning procedure of Section 3.3 (derived from Theorem 3.2), we obtain a simple modification of the Set Covering Machine algorithm that can provably *retrieve* all the causal parents $X_A$ of $Y$. Algorithm 2 shows the pseudocode for learning a conjunction with the proposed Invariant Causal Set Covering Machines (ICSCM) algorithm. The differences with the classical SCM (Algorithm 1) are highlighted in blue.

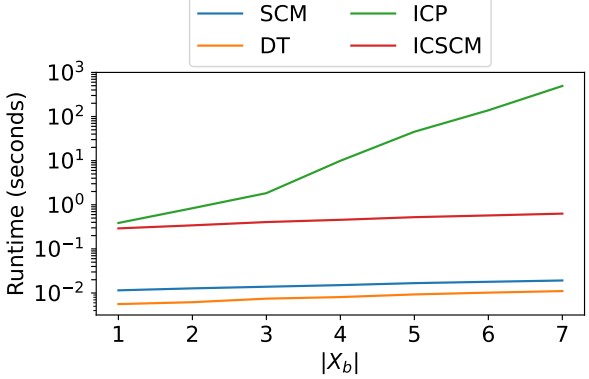

| Model | 1 | 2 | 3 | 4 | 5 | 6 | 7 |
|-------|------|------|------|------|------|------|------|
| SCM | 0.00 | 0.00 | 0.00 | 0.00 | 0.00 | 0.00 | 0.00 |
| DT | 0.00 | 0.00 | 0.00 | 0.00 | 0.00 | 0.00 | 0.00 |
| ICP | 0.96 | 0.98 | 0.99 | 0.99 | 0.97 | 0.09 | 0.00 |
| ICSCM | 0.96 | 0.97 | 0.99 | 0.96 | 0.97 | 0.96 | 0.97 |

Figure 5: Identification of the causal parents $\mathbf{X}_A$ on the simulated data: proportion of 100 training runs where the model relied solely on $\mathbf{X}_A$, for an increasing number of distractor features $\mathbf{X}_B$ (1 to 7). Worst values colored.

Figure 4: Running time for a growing size of of $\mathbf{X}_B$ on simulated data. Additional results in Fig. 9.

Of note, the runtime complexity of ICSCM is $O(m \cdot |\mathcal{R}| \cdot n)$, where $|\mathcal{R}|$ is typically linear w.r.t. $|\mathbf{X}|$. This is in sharp contrast with the exponential runtime complexity of non-linear ICP (Heinze-Deml et al., 2018).

In the experiments presented below, the set $\mathcal{R}$ of candidate rules is defined as:

$$\mathcal{R} = \{\mathbf{1}(x_i > t),\ \mathbf{1}(x_i \leq t) \mid i \in \{1, \ldots, d\},\ t \in \mathcal{T}_i\}$$

where $d$ denotes the dimensionality of $\mathbf{X}$, and $\mathcal{T}_i$ is the set of observed values taken by the variable $X_i$.

## 4 Experiments

We start by conducting experiments in a controlled setting, with simulated data that is guaranteed to satisfy the assumptions of our method (Section 4.1). We then compare the ICSCM to its "non-causal" variant, SCM, on real-world datasets in which our assumptions are unlikely to hold completely (Section 4.2).

### 4.1 Validating the Theory in simulation

**Data:** We parametrize a discrete Bayesian network that satisfies the assumptions at Section 2.1. We define two causal parents $\mathbf{X}_A = [X_{A1}, X_{A2}]$ and a single descendent $\mathbf{X}_C$ for $Y$. We let $\mathbf{X}_B$ be a distractor, unrelated to $Y$ and vary the number of distractors, $|\mathbf{X}_B|$, throughout the experiments. Of particular note, the Bayesian network is designed such that the relationship between $Y$ and $\mathbf{X}_c$ is less noisy than the relationship between $Y$ and its causal parents $\mathbf{X}_A$. As such, algorithms that are vulnerable to spurious associations will tend to use $\mathbf{X}_C$ instead of $\mathbf{X}_A$. In all experiments, we use $m = 10^4$ samples from each of two environments ($E$).

More precisely, we define two environments ($\mathcal{E} = \{0, 1\}$), each one contains $10^4$ observations. For each observation, the causal parent variables $X_{A_1}$ and $X_{A_2}$ are binary-valued randomly generated according to

$$P(X_{A_1}{=}1|E{=}0) = 0.1, \quad P(X_{A_1}{=}1|E{=}1) = 0.5;$$
$$P(X_{A_2}{=}1|E{=}0) = 0.5, \quad P(X_{A_2}{=}1|E{=}1) = 0.3.$$

Note that the environment $E$ affects $\mathbf{X}_A$ by changing the distributions of $X_{A_1}$ and $X_{A_2}$. Conformably to Theorem 2.4, the value of $Y \in \{0, 1\}$ is generated as follows (rules $r_1$ and $r_2$ are identities, and $\epsilon_y = 0.05$):

$$Y = g_y(r_1(\mathbf{X}_A, \epsilon_1) \wedge r_2(\mathbf{X}_A, \epsilon_2), \epsilon_y)$$
$$= g_y(X_{A_1} \wedge X_{A_2}, \epsilon_y)$$
$$= tv + (1-t)X_{A_1} \wedge X_{A_2}, \text{ with } t \sim \mathrm{Ber}(\epsilon_y), v \sim \mathrm{Ber}(0.5)$$

Table 1: Characteristics of the real-world datasets. The first column indicates the number of samples. The second column shows the total number of variables, followed by the number of causal variables among them (in parentheses).

|                     | # samples   | # variables (# causal) |
|---------------------|-------------|------------------------|
| acsfoodstamps       | 840 582     | 239 (137)              |
| acsincome           | 1 664 500   | 232 (66)               |
| assistments         | 2 667 776   | 26 (20)                |
| college scorecard   | 124 699     | 118 (11)               |
| diabetes readmission| 99 493      | 183 (37)               |
| meps                | 26 402      | 3 595 (120)            |

Hence, $Y$ depends only on its causal parents and not directly on $E$.

Then, $X_C = (1-u)Y + uE$, with $u \sim \text{Ber}(0.05)$. This reflects the effect of both $Y$ and $E$ on $\mathbf{X}_C$. Of note, $X_C$ is a better predictor of $Y$ than $X_{A_1} \wedge X_{A_2}$, because even if the noise term $u = 1$, $X_C$ can still have the same value as $Y$ when $E = Y$.

Finally, the variables in $\mathbf{X}_B$ are generated as random variables $X_{B_i} \sim \text{Ber}(\frac{1}{2})_{i=1}^{|\mathbf{X}_B|}$. The experiments are conducted with $|\mathbf{X}_B|$ ranging from 1 to 7.

**Baselines:** The baselines that we consider are Set Covering Machines (SCM; Marchand & Shawe-Taylor (2002)), decision trees (DT; Breiman et al. (1984)), and non-linear ICP (ICP; Heinze-Deml et al. (2018)). ICP should be robust to spurious associations, while DT and SCM should not. See Section A.2 for implementation details.

**Identification of causal parents:** Section 4.1 compares the ability of all methods to identify the causal parents $\mathbf{X}_A$ of $Y$. As expected, SCM and DT always rely on spurious associations and fail to identify $\mathbf{X}_A$. On the contrary, both ICP and ICSCM succeed at identifying $\mathbf{X}_A$ in most cases. The performance of ICP degrades as dimensionality increases, likely due to type II errors that arise due to the vanishing statistical power of its conditional independence tests. In contrast, ICSCM appears less affected by dimensionality; this seems to hold even up to hundreds of variables (see Table 3 in the supplementary material). In Section A.8, we report additional results that support these claims and provide an extensive discussion on the effect of type I and II errors on both methods. Finally, it is clear from these results that ICSCM endows SCM with the ability to identify causal parents.

**Runtime complexity analysis:** Fig. 4 shows the running time of all methods with respect to dimensionality. Clearly, that of ICP increases exponentially fast, while that of ICSCM increases linearly, as expected in Section 3. This makes it more amenable to real-world applications where variables are plentiful, provided, of course, that our functional form assumption (Theorem 2.4) realistically holds.

**Extended simulations experiments:** In additional experiments, we explored the effect of changing the value of the threshold $\alpha$ in the statistical test determining independence in ICSCM. It appear that $\alpha = 0.05$ is within the range of optimal values, the details are presented in section A.3. Another experiment was performed to validate that ICSCM applies to both conjunctive and disjunction causal structures, it is presented in section A.4.

## 4.2 Evaluation on Real-World Data

The previous results support the theoretical soundness of our algorithm. However, they do not assess the method in the wild, where its assumptions are not guaranteed to hold.

**Data:** We, therefore, conduct an empirical study on six real-world datasets from the tableshift benchmark (Hardt & Kim, 2022). For each of these, we have access to a list of variables known to be causes ($\mathbf{X}_A$) of the target label ($Y$), among many other features, as defined in Nastl & Hardt (2024). Moreover, we use Nastl & Hardt (2024)'s notion of "in distribution" and "out of distribution" to define two environments ($E$) for each dataset. Note, however, that we have no guarantee that the hypotheses presented in Section 2.1 and

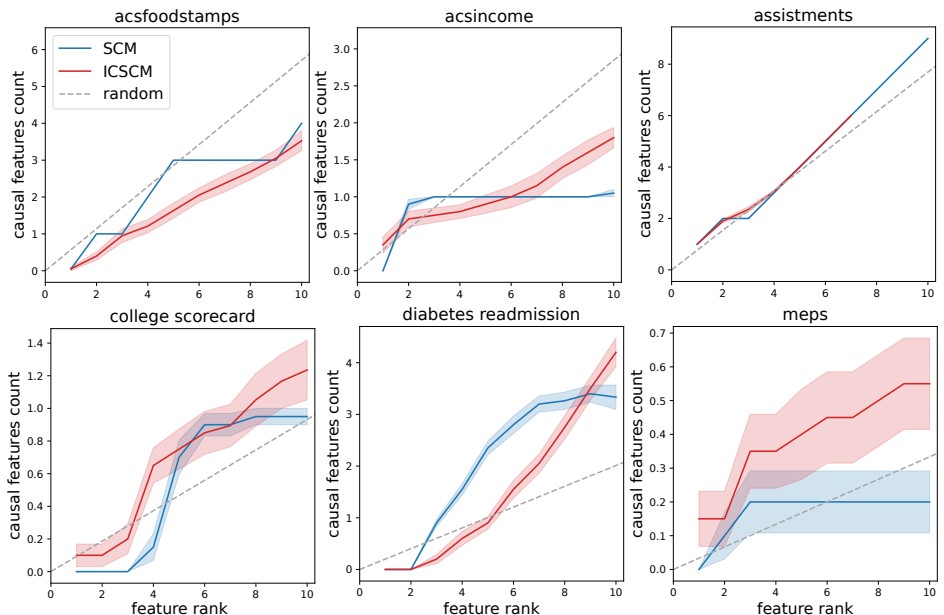

Figure 6: Number of causal features discovered as a function of the model size (feature rank) for real-world datasets. The solid lines give the average causal feature count over 20 repetitions, and the shaded areas report the standard errors. The gray dashed line indicates the expected number of causal variables selected by a random pick. The detailed behavior of each model is shown in supplementary Figures 13 to 18.

Fig. 1 hold, as is common in real-world settings. For example, the relationship between $Y$ and $E$ may be confounded, or the relationship between $\mathbf{X}_A$ and $Y$ may not be a conjunction, among others, which makes for a challenging benchmark.

The datasets, which we now outline, are summarized in Table 1. The "ACS Foodstamps" dataset describes people (U.S. citizens) by socioeconomic variables. The task is to predict Foodstamps recipiency, and the environment is based on geographical location (U.S. census Divisions). The "ACS Income" dataset describes people (U.S. citizens, employed adults) by socioeconomic variables. The task is to predict whether their income is "high" or "low", and the environment is based on geographical location (U.S. census Divisions). The "ASSISTments" dataset describes school problems in an online learning tool and a student attempting to solve them. The task is to predict whether the student solves the problem, and the environment is defined based on the school of the student. The "College scorecard" dataset describes United States colleges. The task is to predict the completion rate of students and the environments are defined based on the type of college (Carnegie classification). The "Diabetes readmission" dataset describes patients with medical information. The task is to predict whether the patients will return to a hospital later; the environment is defined based on the hospital of first admission. The "MEPS" (Medical Expenditure Panel Survey) dataset describes people by socioeconomic variables. The task is to predict the use of health care; the environment is defined by the type of health insurance. Details can be found in Nastl & Hardt (2024).

**Experimental Details:** The candidate rules for the models ($\mathcal{R}$) are thresholds on the value of single features (a.k.a. decision stumps). For each feature, we consider up to 50 thresholds, obtained by quantizing the range of values it takes across the training set. Then, for each feature and threshold, two rules are defined: the first returns "True" if the feature's value is above the threshold and "False" otherwise; the second is the negation of the first. The maximum size of conjunctions is set to $n = 10$. For each dataset, we randomly split the data into two halves—the training and testing sets—and repeat this 20 times.

The set of candidate rules $\mathcal{R}$ must be chosen before running the SCM algorithm and depends on the nature of the data. In the case of Boolean-valued attributes, i.e. $\mathbf{x} \in \{0,1\}^n$ it is natural to define one rule per attribute ($r_k(\mathbf{x}) = x_k$ for $k = 1, \ldots, n$). In the case of continuous real-valued attributes, i.e., $\mathbf{x} \in \mathbb{R}^n$, it is

Table 2: Prediction errors averaged over the 20 splits. The values on the test set are followed by the value on the train set in parentheses. The standard deviations are given in the supplementary Table 4.

|  | SCM | ICSCM |
|---|---|---|
| acsfoodstamps | 0.19 (0.19) | 0.19 (0.19) |
| acsincome | 0.27 (0.27) | 0.32 (0.32) |
| assistments | 0.07 (0.07) | 0.11 (0.11) |
| college scorecard | 0.08 (0.08) | 0.11 (0.11) |
| diabetes readmission | 0.38 (0.38) | 0.46 (0.46) |
| meps | 0.27 (0.26) | 0.34 (0.33) |

common in many learning algorithms (as decision trees) to define rules as decision stumps, that relies on a attribute index $k = 1, \ldots, n$, a threshold values $t \in \mathbf{R}$ and a direction $d \in \{-1, 1\}$ such that $r(\mathbf{x})_{k,t,d} = x_k \geq t$ if $d = 1$ and $r(\mathbf{x})_{k,t,d} = x_k < t$ otherwise. The size of the set $\mathcal{R}$ impacts the computing time linearly.

**Results:** We compare ICSCM to its non-causal counterpart, SCM, based on two factors: (i) the ability to identify causal features and (ii) the prediction error of models.

First, we compare the number of causal features used by each algorithm as a function of model size. To support the ability to identify causal features *better than chance*, we include a random baseline corresponding to the expected number of causal variables that would have been selected by a random pick without replacement. This is simply the expectation of a hypergeometric distribution with parameters derived from Table 1. The results, averaged over 20 trials, are illustrated in Fig. 6. Overall, we observe that, for $n = 10$, ICSCM identifies more causal features than SCM for 4/6 datasets and that both methods are on par for 1/6 datasets. Note that it is not surprising that SCM also relies on causal features sometimes, as these may be good predictors of $Y$. Finally, note that ICSCM identifies more causal features than the random baseline for 4/6 datasets, while the SCM does so for 2/6 datasets. Qualitatively, we observe that ICSCM generally tends to keep selecting causal features as the model size increases, while the number of causal features selected by the SCM could stagnate once a certain count is reached.

Second, we assess the test error of the models learned using SCM and ICSCM. The results, averaged over 20 trials, are presented in Table 2. Overall, SCM tends to build models that are slightly more accurate than those of ICSCM. This is in line with the observations of Hardt & Kim (2022) on the same benchmark datasets, which showed that using only causal variables leads to reduced predictive accuracy, emphasizing a tradeoff between understanding and predicting. It results from the fact that prioritizing invariant rules excludes some rules that contribute to predictive power (e.g., causal children Xc of Y). Yet, the accuracy of ICSCM models does not differ drastically from those of SCM.

Given that real datasets are likely to break the assumptions underlying ICSCM, these results exhibit a good tendency, showing that the modifications proposed to the original SCM algorithms favor the discovery of causal variables. Hence, ICSCM unlocks a deeper understanding of Y which is not possible to achieve with SCM.

## 5 Conclusion

This work introduced ICSCM, a learning algorithm that builds conjunctive and disjunctive models that, in some settings, are guaranteed to rely exclusively on the causal parents of a variable of interest. The algorithm is supported by a rigorous theoretical foundation that exploits invariances that hold in a multi-environment setting to avoid spurious associations. In contrast with previous work, such as non-linear ICP (Heinze-Deml et al., 2018), ICSCM leverages a functional form assumption to considerably reduce the number of statistical tests required, leading to a polynomial-time algorithm that is more amenable to practical applications. Results on simulated data support the validity of the proposed algorithm and theory, and results on real-world datasets suggest that it is better at identifying causal variables than the original SCM algorithm. We envision applications of this method in areas such as biomarker discovery in genomics,

where Set Covering Machines have been shown to perform well. Future work should explore these settings and consider extensions motivated by practical constraints.

**Cautionary note:** The guarantees of the ICSCM are conditional on the functional form and environmental assumptions. The model's outputs should note be taken as conclusion on the causal nature of some variables. They are meant to be interpreted alongside domain experts.

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

# A  Appendix

## A.1  Experimental details

The code for the simulated experiments is available at:
https://github.com/thibgo/icscm-expe-simulated-data.

The code for the experiments on real datasets is available at:
https://github.com/thibgo/icscm-expe-real-data.

## A.2  Baselines and implementation details

For the Set Covering Machine (SCM) and classification tree (DT) baselines, as well as ICSCM, we conducted a hyperparameter search using 5-fold cross-validation and selected the values that led to the highest binary accuracy, on average, over all folds. Details are provided below:

- Set Covering Machine (SCM; Marchand & Shawe-Taylor (2002)): We used the implementation available at https://github.com/aldro61/pyscm (version 1.1.0) and considered the following hyper-parameter values for trade-off $p$: $\{0.1, 0.5, 0.75, 1.0, 2.5, 5, 10\}$. The models built were conjunctions.

- Classification trees (DT; Breiman et al. (1984)): We used the implementation available in Scikit-Learn (version 1.2.2; Pedregosa et al. (2011)) and considered the following hyperparameter values: i) maximum depth: $\{1, 2, 3, 4, 5, 10\}$, ii) minimum samples split: $\{2, 0.01, 0.05, 0.1, 0.3\}$.

- ICSCM: We implemented the pseudo-code showed in Algorithm 2 in Python and considered the following hyperparameter values for utility $p$: $\{0.1, 0.5, 0.75, 1.0, 2.5, 5, 10\}$. The models built were conjunctions. Conditional independence was tested using a $\chi^2$ test with $\alpha = 0.05$. For the pruning procedure, we used the conditional G-test implementation available at https://github.com/keiichishima/gsq, also with $\alpha = 0.05$.

For the ICP baseline, we reimplemented the method in Heinze-Deml et al. (2018). The implementation is available in our main codebase. Conditional independence was tested using a conditional G test with $\alpha = 0.05$. We used the conditional G-test implementation available at https://github.com/keiichishima/gsq.

## A.3  Evaluating the effect of the test threshold alpha

The threshold $\alpha$ is one of the hyperparameters of ICSCM. It defines the level of significance of the test required to consider independance between . To investigate its influence, we repeat the simulated data experiment with various values of the threshold $\alpha$. The results heatmap (Fig. 7) shows the proportion of 100 training runs where ICSCM correctly identified the causal parents $\mathbf{X}_A$, for an increasing number of distractor features $\mathbf{X}_B$ (1 to 7). The value of $\alpha$ varies from 0.001 to 1. Observe that the results for $\alpha = 0.05$ (the default value used for ICSCM) correspond to those in Section 4.1. We make the following observations:

- As alpha decreases, the false negative (type 2 error) rate of the statistical tests increases. This means that some conditional dependencies, created by conditioning on rules that refer to variables in $\mathbf{X}_C$, are undetected. The consequence is a higher propensity to include non-causal parents in the model.

- As alpha increases, the false positive (type 1 error) rate of the statistical tests increases. This means that some rules that refer to causal parents are incorrectly discarded since they are found to lead to "hallucinated" conditional dependencies.

- Finally, we note that there is a wide range of alpha values where the effect on the identification of causal parents is minimal. Our recommended default value of $\alpha = 0.05$ is right within this range.

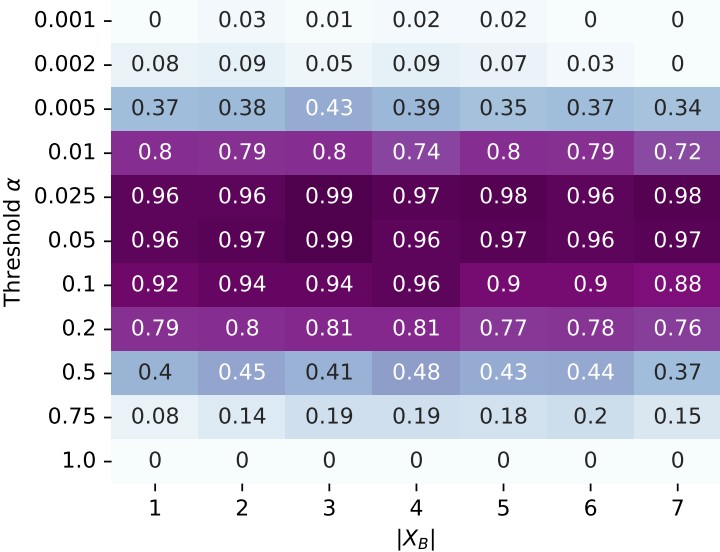

Figure 7: Rate of correct identification of causal parents by ICSCM, given different values of the independence test threshold $\alpha$.

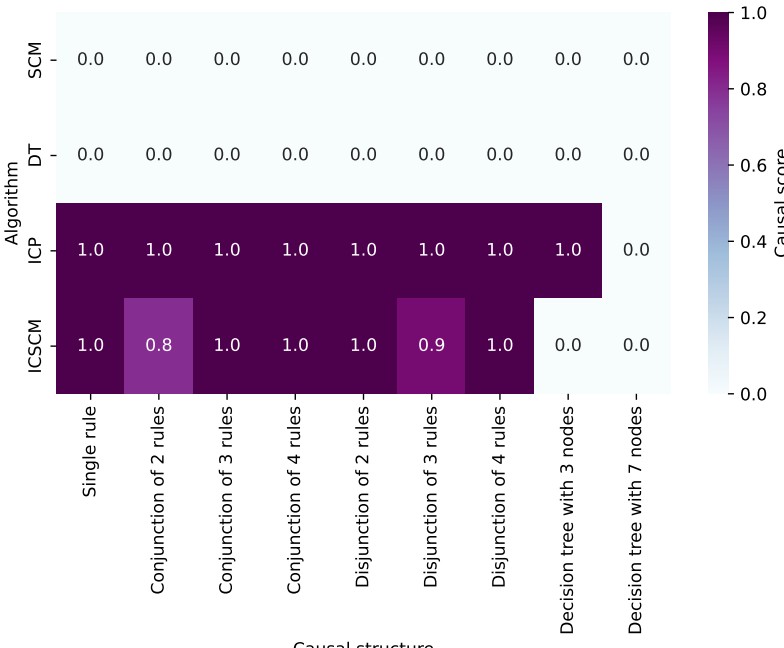

Figure 8: Score (causal set identification rate) of algorithms on different causal structures. Note that ICSCM failure on decision trees is expected, since assumption 2.4 is not satisfied.

## A.4 Evaluating algorithms behavior on difference causal structures

The ICSCM is presented in its conjunctive form, but symmetrically, it is also able to learn disjunctive causal relationships. This statement is confirmed by the results presented in Fig. 8. It shows the proportion of cases, over 100 trials, where ICSCM and other algorithms properly identified the causal parents for various

Table 3: Identification of the causal parents $\mathbf{X}_A$: proportion of 100 training runs where the model relied solely on $\mathbf{X}_A$, for an increasing number of distractor features $\mathbf{X}_B$ (1 to 200).

| $|\mathbf{X}_B|$ | 1 | 2 | 3 | 4 | 5 | 6 | 7 | 10 | 15 | 20 | 25 | 50 | 100 | 200 |
|---|---|---|---|---|---|---|---|---|---|---|---|---|---|---|
| ICSCM | 0.96 | 0.97 | 0.99 | 0.96 | 0.97 | 0.96 | 0.96 | 0.89 | 0.89 | 0.86 | 0.96 | 0.90 | 0.91 | 0.94 |

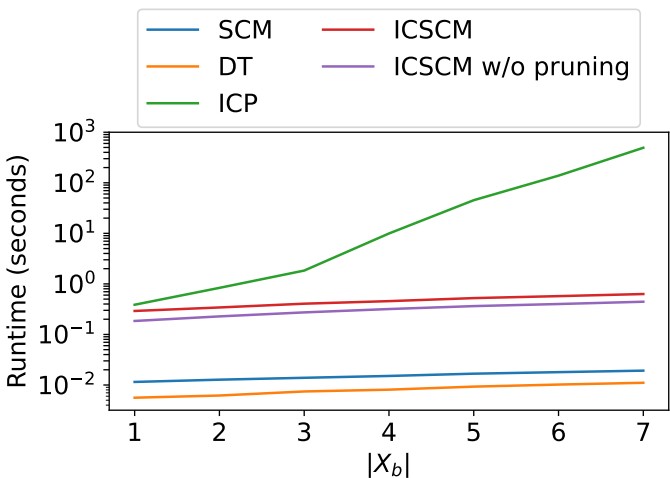

Figure 9: Running time of algorithms. The ICSCM is showed in two versions : the normal with pruning and a version without the pruning phase (see Section 3.3).

kinds of causal relationships (conjunctions, disjunctions). Here, we vary the size of $X_A$ from 1 to 4 variables, set the number of variables in $X_B$ to 3 variables, and the size of $X_C$ to one variable. These new results clearly show that ICSCM works well in the conjunctive and disjunctive cases. Furthermore, for completeness, we include cases where the true labeling mechanisms are decision trees, in direct violation of the assumptions required by ICSCM. As expected, the performance is not on par with cases where our assumptions were met. It is the very nature of the SCM to build conjunctions and disjunctions of binary rules, which is both a limitation and an asset, as this commitment allows the algorithm to rely on an efficient greedy building mechanism.

ICP is able to retrieve the set of causal variables that forms a tree of size 3. But it fails on the similar setting with 7 causal variables. This may be explained by the number of variables rather than the structure. The number of statistical tests grow exponentially with the number of variables in the problem, and so grow the probability of an error in one of the tests. We discuss this in the next Section A.8.

### A.5 Evaluating the effect of a very large number of distractor variables

The simulated experiments in section 4.1 explore the effect of a growing number of distractor variables in $\mathbf{X}_B$. They are limited to $|\mathbf{X}_B| = 7$ since the runtime of ICP increases exponentially with the total number of variables in the problem. In Table 3 we explore the behavior of ICSCM for $|\mathbf{X}_B|$ up to 200. It appear that ICSCM continues to robustly learn causal models in this context.

### A.6 Computing cost of the pruning phase of the ICSCM

The pruning phase is the ultimate step of the ICSCM algorithm (2). It is necessary to guarantee that the model does not contains noise variables, as shown in Section 3.3. In Fig. 9, we highlight the computing time of this phase. It appear that the pruning does not influence much the total ICSCM runtime, and it is compatible with the objective of running in polynomial time.

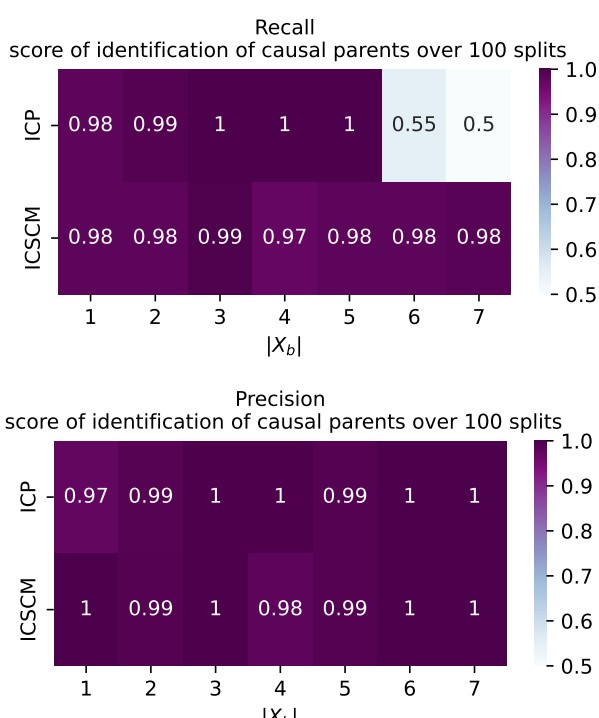

Figure 10: Precision and recall metrics on the task of identifying the set of causal parents for ICP and ICSCM. The scores are computed for several values of $|\mathbf{X}_B|$, and averaged over 100 randomly generated datasets, using the experimental design presented in Section 4.1.

## A.7 Stopping criterion effect on real data

On the simulated data experiments of section 4.1, the stopping criterion applies as expected. The model stops learning when there is independence between $E$ and $Y$ among the samples of the final leaf. When we did not constrain the maximum number of rules in the real-data experiments of section 4.2, we observed that the ICSCM had a tendency to select a large number of rules, often more than one hundred. The sample size in the statistical test used to obtain p-value $\gamma$ does not decrease that fast. This phenomenon happens because this statistical test is performed using all "non-final" samples $\mathcal{P} \cup \mathcal{N}$, which are the samples classified as positive by the conjunction so far (both true positives and false positives). In other words, seeing the conjunction as a binary tree (as in Fig. 2), the ICSCM grows a heavily unbalanced tree, propagating all the positively classified samples to the always unique positive leaf. This is in sharp contrast with regular decision trees, where the samples are distributed "more evenly" among multiple positive and negative leaves. In the reported real-data experiments, we fixed a small number of maximum rules, hence focusing on the 10 invariant rules having the best utility scores and interest in causal discovery.

## A.8 Robustness to Type I and Type II Errors

While algorithms like ICP and ICSCM offer strong theoretical guarantees, their empirical performance is subject to the reliability of the (conditional) independence tests that they perform. Empirical phenomena such as *type I* errors (false positive: finding dependence when there is none) and *type II* errors (false negative: finding independence when there is dependence) can affect their performance in practice. Type I errors can be controlled via the $\alpha$ threshold on the p-value of the statistical tests. However, type II errors are more difficult to control, as the false negative rate $\beta$ depends on the statistical power of a test, which in turn depends on factors such as the effect size (which we do not control) and the number of available data samples. Here, we discuss the effect of each type of error on both algorithms.

**ICP:**      To find the set of causal parents $\mathbf{X}_A$, ICP tests the independence of $Y$ and $E$ conditioned on every possible set of variables, resulting in $2^{|\mathbf{X}|}$ conditional independence tests. Then, the algorithm takes the intersection of all sets that were found to lead to independence (see Heinze-Deml et al. (2018), line 4 of Algorithm 1 in Appendix B). The intersection has an effect similar to the pruning procedure of ICSCM (see Theorem 3.2) and serves to filter out $\mathbf{X}_B$ from the solution. ICP is quite robust to type I errors since such errors result in discarding only a few of the sets that contain $\mathbf{X}_A$. Apart from rare cases, such as making a type I error for the set $\{\mathbf{X}_A\}$ and not for all sets containing both $\mathbf{X}_A$ and $\mathbf{X}_B$, the intersection renders the algorithm robust to type I errors. On the other hand, ICP is vulnerable to type II errors, which can cause it to incorrectly include a set that does not lead to independence, e.g., $\{\mathbf{X}_C\}$ or $\{X_{A1}\}$ (but missing some $X_{Ai} \in \mathbf{X}_A$) in the intersection. If this happens, the intersection returns either the empty set or a partial set of causal parents. As the dimensionality of $\mathbf{X}$ increases (for a fixed sample size), the power of the statistical tests decreases and the probability $\beta$ of making such errors increases, amplifying the problem. We hypothesize that this is what causes the poor performance of ICP for the larger sizes of $\mathbf{X}_B$ at Section 4.1. This is supported by the results at Fig. 10, which show that the *recall* of ICP on the task of identifying the set of causal parents drops for $|\mathbf{X}_B| \geq 6$. In contrast, note that the *precision* remains stable (see Fig. 10), illustrating the robustness of this method to type I errors.

**ICSCM:**      We separate the discussion of how ICSCM can be affected by type I and II errors into three parts, based on the different stages of the algorithm. For simplicity, let us assume that the set of candidate rules $\mathcal{R}$ contains a single rule per causal parent.

1. **Effect on Eq. (2):**      This criterion is used to select which rules are permitted to be added to the model. A type I error has the effect of rejecting a rule that actually satisfies the criterion, e.g., rejecting a causal parent in $\mathbf{X}_A$. If this happens, the resulting conjunction might not contain all the causal parents. However, note that, even if a causal parent is rejected at one stage of building the conjunction, the data filtering that occurs at the end of every iteration in Algorithm 2 results in re-testing the same rule with a subset of the data at the next iteration, which might offer some resistance to type I errors. As for type II errors, these correspond to incorrectly believing that the criterion is satisfied and could result in adding rules that depend on $\mathbf{X}_C$ to the conjunction. In both cases, if such errors were to be made, the stopping criterion at Eq. (3) would not be satisfiable due to unblocked paths between $Y$ and $E$ in $G$. In terms of precision and recall w.r.t. the causal parents, type I and type II errors would result in lower recall and precision, respectively.

2. **Effect on Eq. (3):**      This criterion is used to determine when to stop adding rules to the model. Here, a type I error would result in a failure to stop. However, note that, even if the algorithm failed to stop, as long as all $R_i$ have been added to the conjunction, the algorithm should find that no other $R_j$ satisfies Eq. (2) and stop, offering some robustness to such errors. As for type II errors, these would result in incorrectly concluding that the criterion is satisfied, resulting in premature stopping and missing causal parents in $\mathbf{X}_A$. Hence, type I and type II errors would result in lower precision and recall, respectively.

3. **Effect on Theorem 3.2:**      This result is the foundation for the pruning procedure of ICSCM. Here, a type I error would result in incorrectly keeping a variable in $\mathbf{X}_B$ in the conjunction instead of pruning it. In contrast, a type II error would result in incorrectly removing a variable in $\mathbf{X}_A$ from the conjunction. Type I and type II errors would therefore result in lower precision and recall, respectively.

Finally, note that while we typically cannot control all the factors that govern type II errors ($\beta$), there is one key element that distinguishes ICSCM from ICP: ICP must conduct conditional independence testing for every possible set of parents, resulting in large conditioning sets. In contrast, ICSCM's tests are only conditioned on a number of variables that grows linearly with the length of the conjunction. As such, ICSCM's tests may have greater power and the algorithm may be less affected by type II errors. The results at Fig. 10, show that both the precision and recall of ICSCM remain high as the number of variables increases, in contrast with ICP whose recall decreases significantly.

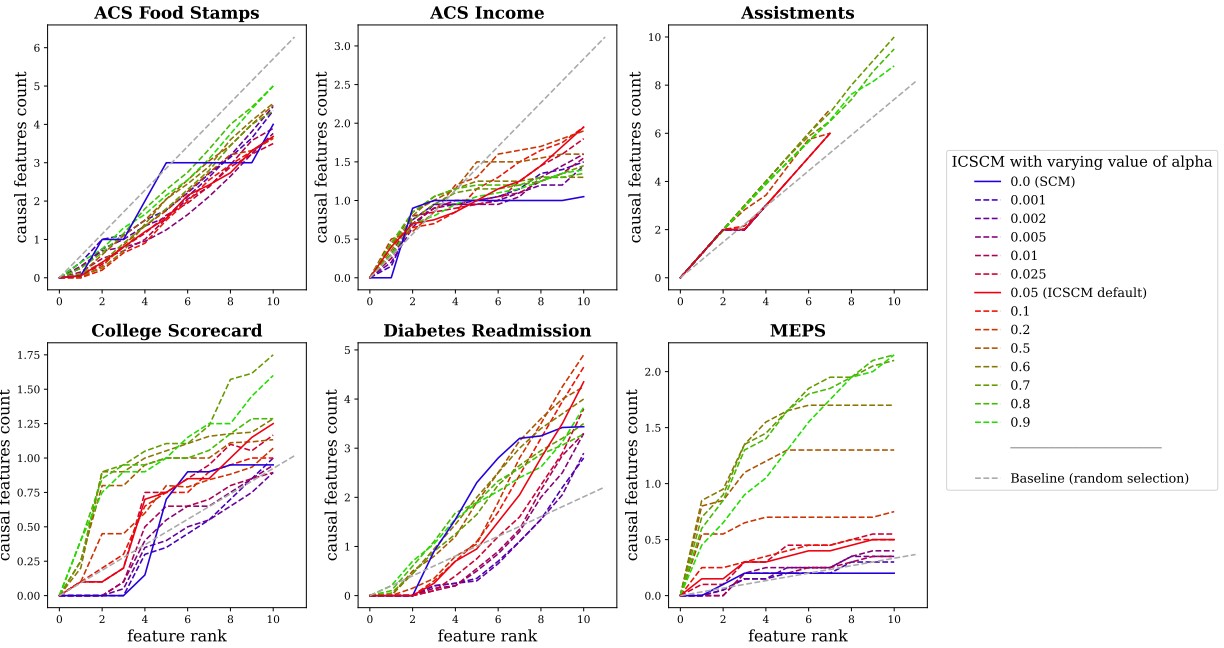

Figure 11: Causal scores of ICSCM models for varying value of hyperparameter $\alpha$.

## A.9 Influence of the hyperparameter $\alpha$

In this section, we present a detailed experiment on the influence of hyperparameter $\alpha$. It is the threshold on the p-value of the statistical independence test as shown in Algorithm 2. Figure 11 shows increasing the value of $\alpha$ tends to moderately increase the number of causal features retrieved by ICSCM. The tendency is more affirmed in the datasets College Scorecard and MEPS, but the overall number of variables retrieved on these datasets remains small, less than 2 causal variables in the first 10 rules. Additionally, Figure 12 shows that increasing $\alpha$ decrease the accuracy in prediction.

When $\alpha$ grows, more rules that create a dependence between $E$ and $Y$ are eliminated. This is expected to positively affect the causal score (i.e., the number of causal features retrieved). Additionally, more rules are mistakenly eliminated because they appear to create a dependence between $E$ and $Y$ while in the ground truth they do not. This is expected to negatively affect the predictive score (i.e., the accuracy).

Therefore, the appropriate value of $\alpha$ depends on the dimensions of the data and the proportion of causal variables. The practitioner should experiment several values of this hyperparameter using their own data.

## A.10 Detailed results of real-world data experiments

The Figures 13 to 18 show the 20 individual curves of SCM and ICSCM that are averaged in Fig. 6. The Table 4 extends the results of Table 2 by including the standard deviation over 20 repetitions.

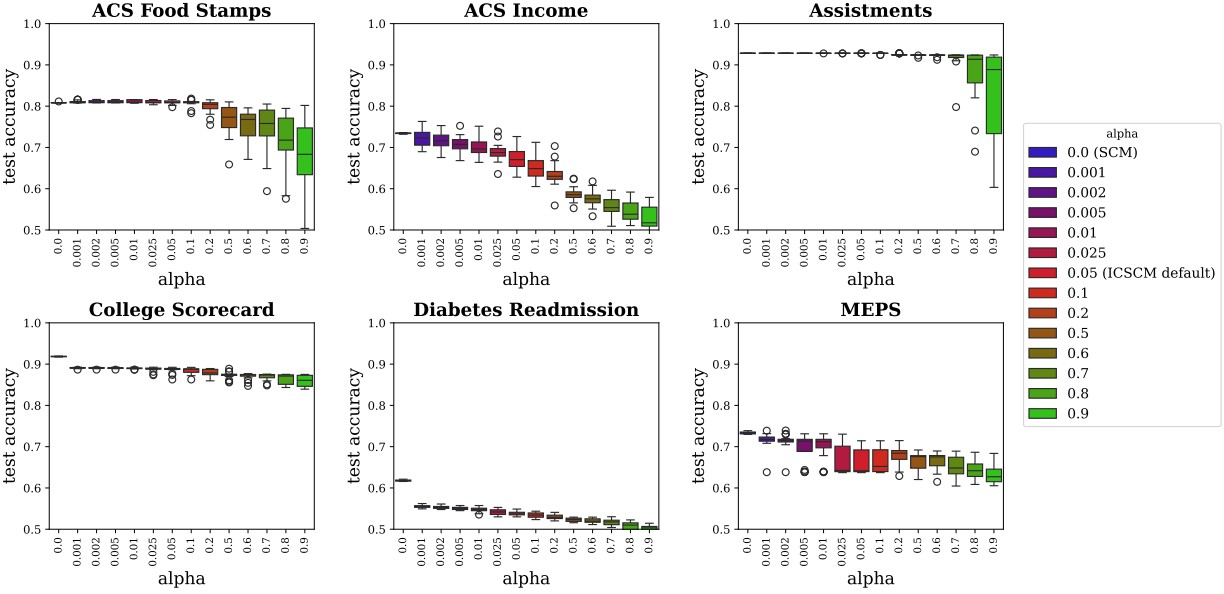

Figure 12: Accuracy scores of ICSCM models for varying value of hyperparameter $\alpha$.

Table 4: Prediction error of the models averaged on the 20 splits. The standard deviation is indicated in parentheses.

| | Train | | Test | |
|---|---|---|---|---|
| | SCM | ICSCM | SCM | ICSCM |
| acsfoodstamps | 0.19 (0.0017) | 0.19 (0.0050) | 0.19 (0.0007) | 0.19 (0.0044) |
| acsincome | 0.27 (0.0013) | 0.32 (0.0222) | 0.27 (0.0010) | 0.32 (0.0225) |
| assistments | 0.07 (0.0005) | 0.11 (0.0299) | 0.07 (0.0001) | 0.11 (0.0298) |
| college scorecard | 0.08 (0.0006) | 0.11 (0.0060) | 0.08 (0.0007) | 0.11 (0.0059) |
| diabetes readmission | 0.38 (0.0016) | 0.46 (0.0058) | 0.38 (0.0015) | 0.46 (0.0052) |
| meps | 0.26 (0.0028) | 0.33 (0.0357) | 0.27 (0.0030) | 0.34 (0.0328) |

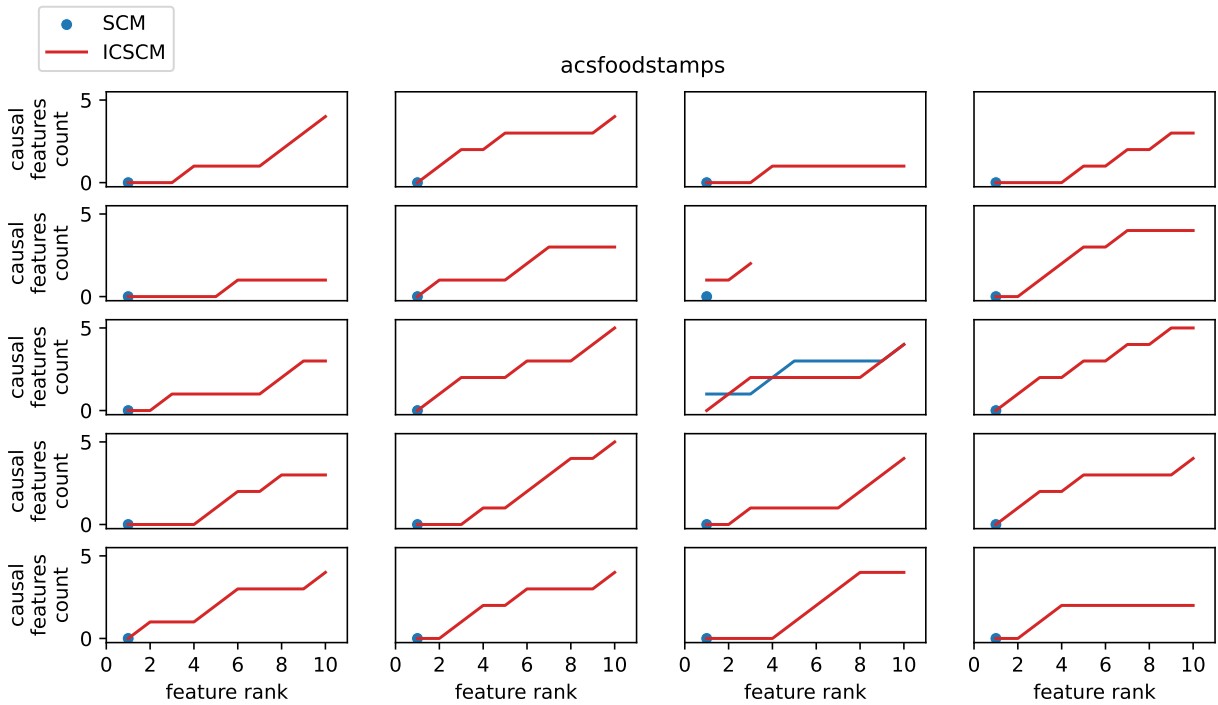

Figure 13: Detailed behavior of the models on each split, on the dataset **acsfoodstamps**, showing the number of causal variables selected.

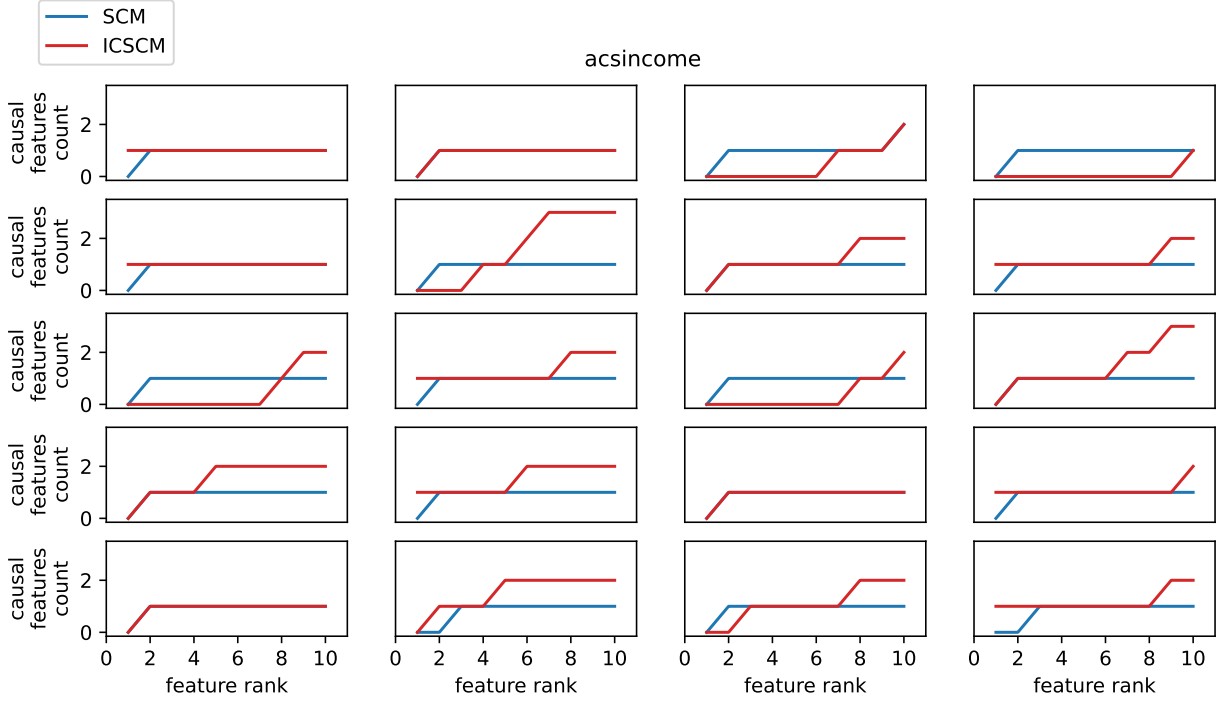

Figure 14: Detailed behavior of the models on each split, on the dataset **acsincome**, showing the number of causal variables selected.

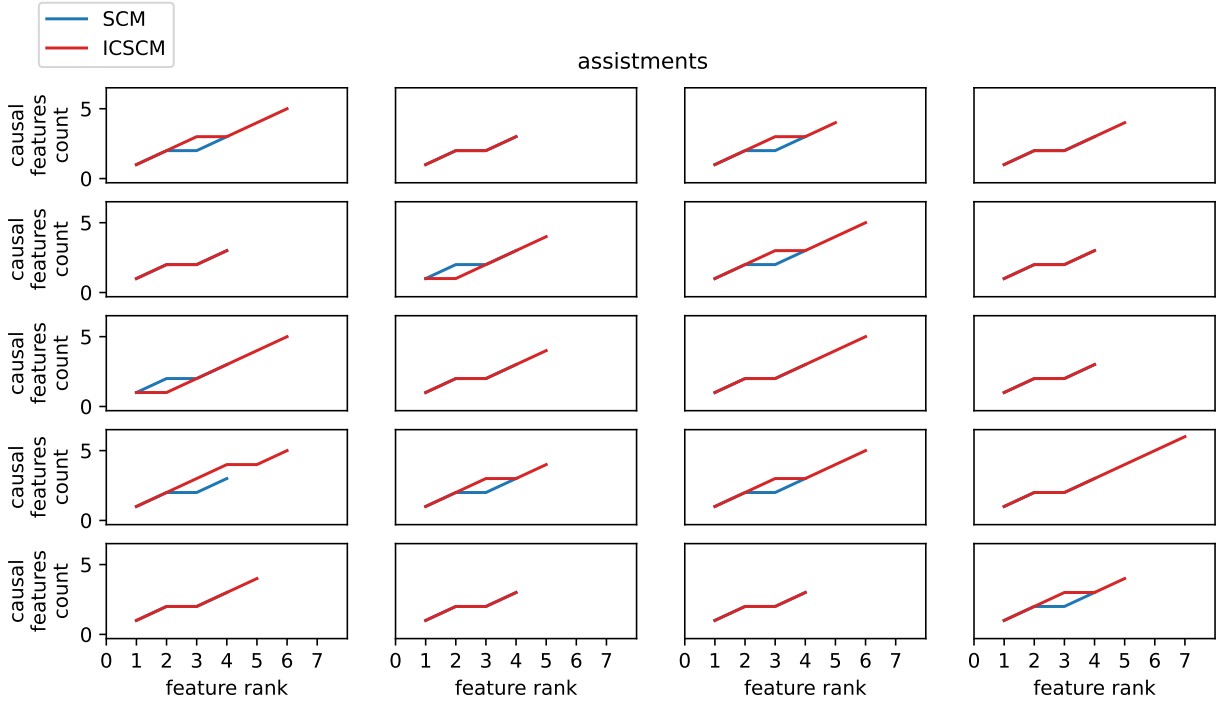

Figure 15: Detailed behavior of the models on each split, on the dataset **assistments**, showing the number of causal variables selected.

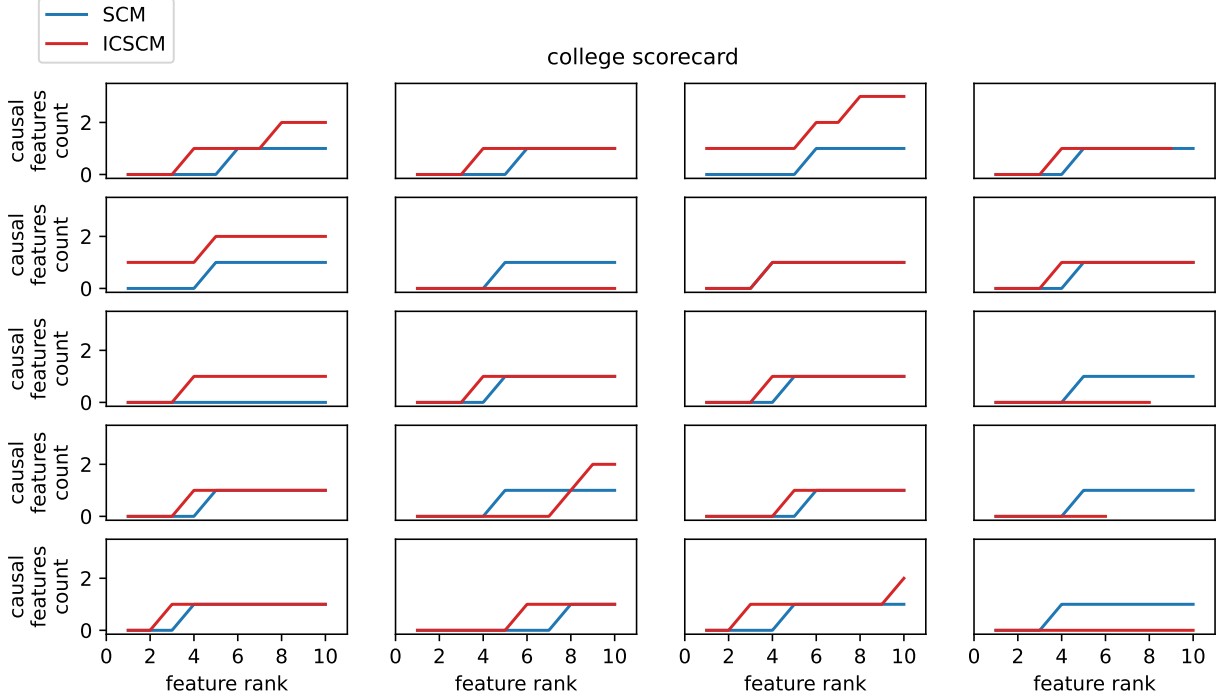

Figure 16: Detailed behavior of the models on each split, on the dataset **college-scorecard**, showing the number of causal variables selected.

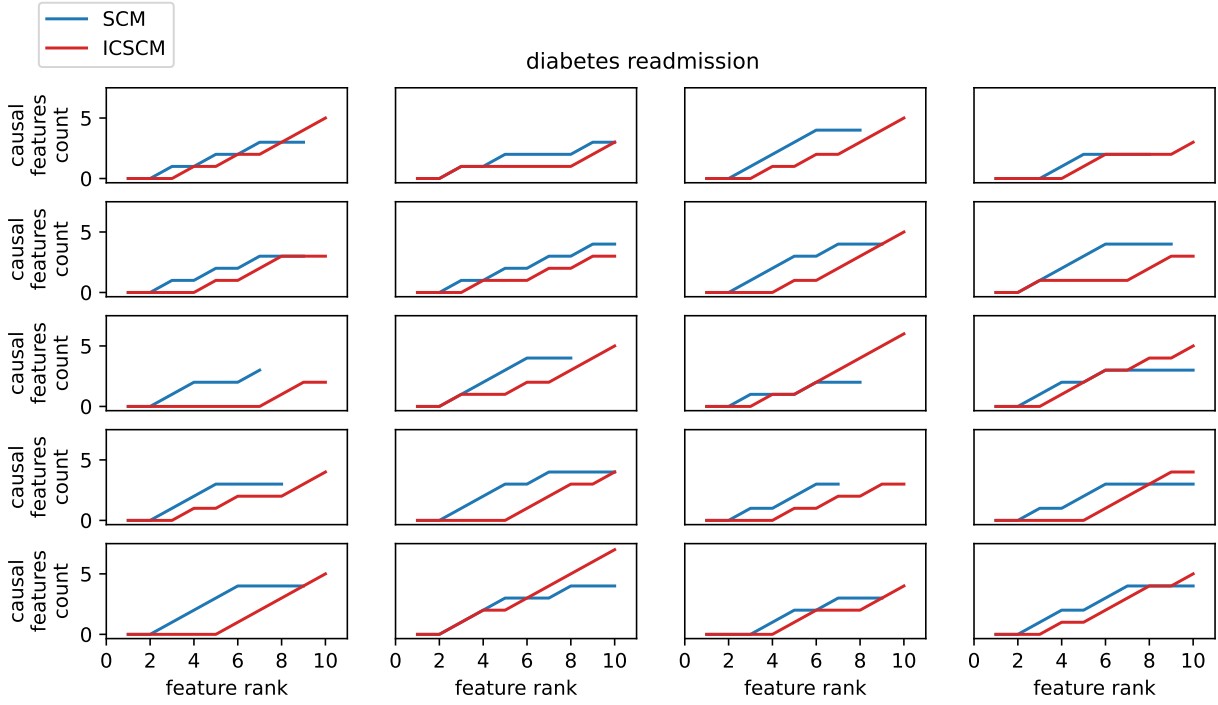

Figure 17: Detailed behavior of the models on each split, on the dataset **diabetes-readmission**, showing the number of causal variables selected.

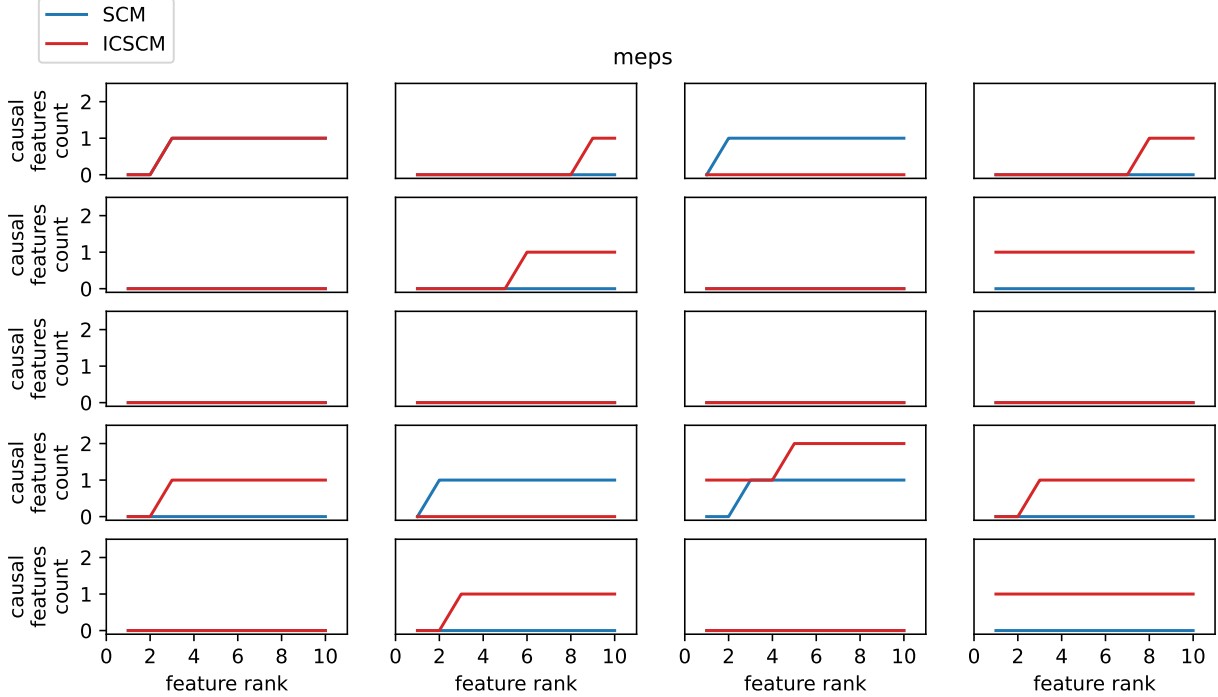

Figure 18: Detailed behavior of the models on each split, on the dataset **meps**, showing the number of causal variables selected.

