# OpenReview forum: "Invariant Causal Set Covering Machine"
_TMLR — Decision pending for TMLR_

### Review · Reviewer_D6h4 · 2026-03-10

**Summary Of Contributions:**

This paper proposes the Invariant Causal Set Covering Machine (ICSCM), an extension of the classical Set Covering Machine (SCM) algorithm that incorporates ideas from invariant causal prediction (ICP) to avoid spurious associations in rule-based classifiers. The key insight is that, under a conjunctive functional form assumption on the data-generating process, the causal parents of a target variable can be identified by enforcing statistical independence between Y and the environment E at each leaf of the conjunction tree. This yields a polynomial-time algorithm (O(m·|R|·n)) that contrasts favorably with the exponential complexity of non-linear ICP. The paper provides a theoretical guarantee (Theorem 3.1) that the proposed construction criteria are sufficient to recover the causal parents, along with a pruning procedure (Proposition 3.2) to remove spurious non-causal variables. Experiments on simulated data validate the theory, and experiments on six real-world datasets from the TableShift benchmark show that ICSCM tends to identify more causal features than vanilla SCM.

**Audience:**

Yes

**Audience Explanation:**

The intersection of interpretable machine learning and causal inference is an active and important research area. Practitioners who use rule-based models (e.g., in genomics, clinical decision support) and wish to extract causally meaningful insights will find this work directly relevant. The polynomial-time guarantee for causal parent identification under conjunctive structure is a concrete and useful result. The paper also provides a clear demonstration of the tradeoff between predictive accuracy and causal identifiability, which is of broader interest to the TMLR community.

**Broader Impact Concerns:**

The primary application domain mentioned is biomarker discovery in genomics and healthcare. In these settings, falsely identified causal variables could potentially mislead downstream drug development or clinical decision-making. A brief cautionary note acknowledging that ICSCM's guarantees are conditional on the functional form and environmental assumptions would be helpful, and that outputs are best interpreted alongside domain expertise. This would help practitioners who may not be fully aware of the underlying assumptions use the method more responsibly.

Regarding the conjunctive functional form assumption (Assumption 2.4), I appreciate that the authors acknowledge its role as both a limitation and an enabler of the method's efficiency. That said, I am curious about the potential for generalization. Real-world causal mechanisms are often continuous or nonlinear, and it would be interesting to understand whether the core invariance-based criteria of Theorem 3.1 could be extended beyond the conjunction/disjunction family. For instance, could similar construction criteria be formulated in terms of more general graphical conditions, such as properties of the Markov blanket of Y? I recognize this may come at a cost to computational tractability; a brief discussion of the authors' intuitions on this direction would enrich the paper.

**Claims And Evidence:**

Yes

**Claims Explanation:**

The theoretical claims are formally proven and the proofs are correct to the best of my assessment. The simulation study is well-designed and directly validates Theorem 3.1 — in particular, the deliberate construction of XC as a better predictor than XA makes the comparison meaningful. The runtime comparison (Figure 4) clearly demonstrates the polynomial vs. exponential gap between ICSCM and ICP. The real-world experiments, while limited by the fact that the underlying assumptions are unlikely to hold exactly, are conducted on a standard benchmark and provide honest reporting of both causal identification and prediction accuracy. Overall, the evidence is appropriate for the claims being made, provided one accepts the functional form assumption.

**Requested Changes:**

1. The sensitivity analysis of α (Figure 7) is useful but the guidance for practitioners on how to choose α in the absence of ground truth causal labels is missing. A brief discussion would be helpful.

2. The real-world evaluation relies on the definition of "causal variables" from Nastl & Hardt (2024). The authors should acknowledge that this ground truth is itself uncertain and discuss how noise in these labels might affect the conclusions.

3. The pruning procedure assumes that no rule jointly involves both XA and XB variables. This assumption is stated but its practical implications for continuous features with decision stumps deserve more discussion.

---

> ### Author Response · Authors · 2026-06-17
>
> We thank the reviewer for the attentive reading and valuable comments on our work.
>
> Following the reviews, we uploaded a revised version of the manuscript. Here is a version of it where the changes are colored in red : https://anonymous.4open.science/r/icscm-expe-real-data-4328/ICSCM___TMLR___changes_after_reviews.pdf
>
> ---
>
> >The sensitivity analysis of α (Figure 7) is useful but the guidance for practitioners on how to choose α in the absence of ground truth causal labels is missing. A brief discussion would be helpful.
>
> To extend the discussion on this parameter, we added an experiment where we vary the value of alpha while building ICSCM models on the 6 datasets from the tableshift benchmark (Section 4.2). This completes the experiments on alpha for simulated data presented by Figure 7 of the appendix. The results are presented in Appendix A.9 of the revised version (they can also be seen here : https://anonymous.4open.science/r/icscm-expe-real-data-4328/figures/figure-alpha-exp.pdf and https://anonymous.4open.science/r/icscm-expe-real-data-4328/figures/figure-alpha-exp-accuracy.pdf). Note that the value $\alpha=0$ corresponds to the SCM algorithm (every candidate rule succeeded the independence test). We observe that $\alpha$ affects the ICSCM's ability to retrieve causal features in 5 of the 6 datasets. In two datasets, high values of $\alpha$ significantly improve the causal score (i.e., the number of causal features retrieved).
>
> When $\alpha$ grows:
> * More rules that create a dependence between $E$ and $Y$ are eliminated. This is expected to positively affect the causal score (i.e., the number of causal features retrieved, as illustrated by the new Figure 11).
> * More rules are mistakenly eliminated because they appear to create a dependence between $E$ and $Y$ while in the ground truth they do not. This is expected to negatively affect the predictive score (i.e., the accuracy, as illustrated by the new Figure 12).
> \end{itemize}
>
> Therefore, the appropriate value of $\alpha$ depends on the dimensions of the data and the proportion of causal variables. The practitioner should experiment with several values of this hyperparameter using their own data.
>
> ---
>
> > The real-world evaluation relies on the definition of "causal variables" from Nastl & Hardt (2024). The authors should acknowledge that this ground truth is itself uncertain and discuss how noise in these labels might affect the conclusions.
>
> We acknowledge that the classification of “causal” and “non causal” variables in observational data involves some subjectivity. Therefore, one should not overanalyse such results. Caution remains necessary as ground truth knowledge is difficult to access in causal data science.
>
> ---
>
> > The pruning procedure assumes that no rule jointly involves both XA and XB variables. This assumption is stated but its practical implications for continuous features with decision stumps deserve more discussion.
>
> Yes, thank you for pointing this out. We have to state that decision stumps are built on a single variable. It is necessary for the pruning, and also for the validity of the statement :
>
>     The runtime complexity of ICSCM is O(m · |R| · n), where |R| is typically linear w.r.t. |X|
>
> We added this point explicitly in Section 3.1. We also mentioned in Section 3.4 how the rules set $R$ is built in the implementation used for the experiments (following the assumption).
>
> ---
>
> > The primary application domain mentioned is biomarker discovery in genomics and healthcare. In these settings, falsely identified causal variables could potentially mislead downstream drug development or clinical decision-making. A brief cautionary note acknowledging that ICSCM's guarantees are conditional on the functional form and environmental assumptions would be helpful, and that outputs are best interpreted alongside domain expertise. This would help practitioners who may not be fully aware of the underlying assumptions use the method more responsibly.
>
> Thank you for suggesting this. We added a “cautionary note” section to remind the reader of the points you mentioned.

---

### Review · Reviewer_KScw · 2026-03-24

**Summary Of Contributions:**

This paper proposes Invariant Causal Set Covering Machines (ICSCM), an extension of the classical Set Covering Machine (SCM) that integrates ideas from invariant causal prediction to mitigate reliance on spurious correlations.

The main contributions include:
- A novel algorithm (ICSCM) that augments SCM with causal invariance constraints.
- A theoretical result establishing conditions under which causal parents can be identified in polynomial time, contrasting with exponential complexity in prior ICP approaches.
- A practical algorithmic framework using independence tests for rule selection, stopping criteria and a pruning procedure to remove non-causal features.
- The authors validate the method on both synthetic data and real-world datasets, and the results show that it can successfully recover causal parents under controlled settings and tends to identify more causal variables than SCM in practice.

**Audience:**

Yes

**Audience Explanation:**

This work appeals to multiple segments of the TMLR audience: causal ML researchers, the interpretable ML community, applied fields like healthcare and genomics.  However, its impact is constrained by strong assumptions that limit applicability.

**Claims And Evidence:**

Yes

**Claims Explanation:**

- The paper provides a convincing and rigorous theoretical proof under its stated assumptions. In particular, the core Theorem 3.1 is clear stated and supports its claim.
- The pseudocode for the ICSCM algorithm is clean, well-annotated, and they also provided the implementation code. Both make the method straightforward to understand and easy to reproduce.
- The simulation experiments are well-aligned with the theory: data is generated based on their assumptions; ICSCM can succesffully recover causal parents; runtime of ICSCM is polynomial compared with the exponential runtime of ICP.
- The performance of ICSCM on real-world datasets is generally in line with expectations, and there are reasonable explanations for the higher test error as well.

**Requested Changes:**

- The algorithm makes very strong assumptions, which should severely limit the scenarios in which it can be used. Could the authors discuss or provide some practical guidance on what real-world situations or what kinds of data this algorithm can be applied to?
- The algorithm relies heavily on statistical independence tests. Therefore, the authors should provide more comprehensive robustness analysis and offer clear guidance on how to select the appropriate tests or parameters under different conditions.

---

> ### Author Response · Authors · 2026-06-17
>
> We thank the reviewer for the attentive reading and valuable comments on our work.
>
> Following the reviews, we uploaded a revised version of the manuscript. Here is a version of it where the changes are colored in red : https://anonymous.4open.science/r/icscm-expe-real-data-4328/ICSCM___TMLR___changes_after_reviews.pdf
>
> ---
>
> > The algorithm makes very strong assumptions, which should severely limit the scenarios in which it can be used. Could the authors discuss or provide some practical guidance on what real-world situations or what kinds of data this algorithm can be applied to?
>
> As the reviewer mentioned, the strong assumptions about the functional form limit the method's scope. That being said, Genomics-and Omics data in general- offers contexts where rule-based models successfully extract the information on the link between the observed variables ($x$) and the variable of interest ($y$). [1]
> In other domains, the suitability of SCMs is also context-dependent. Nevertheless, implementations of both the SCM and ICSCM algorithms are publicly available and can be evaluated without any domain-specific prior knowledge. Whether the model performs adequately relative to competing approaches will become apparent rapidly through standard predictive evaluation. Should the SCM prove to be a strong predictive model in a given setting, the application of ICSCM is then well-motivated and methodologically sound.
> It is worth noting that several studies have reported competitive performance of SCMs on real-world datasets [2]. This is consistent with the fact that SCMs constitute a special case of decision tree models, a family of methods whose appropriateness across a broad range of applied settings is well established in the machine learning literature. In the absence of prior knowledge of an SCM's suitability for a given dataset, a natural and practical approach is to fit the model and assess its adequacy using predictive performance metrics, with standard model comparison procedures as a diagnostic tool.
> [1] Godon, T., Plante, P., Corbeil, J., Germain, P., & Drouin, A. (2025). On Selecting Robust Approaches for Learning Predictive Biomarkers in Metabolomics Data Sets. Analytical chemistry.
> [2] Marchand, Mario and John Shawe-Taylor. “The Set Covering Machine.” J. Mach. Learn. Res. 3 (2003): 723-746.
>
> ---
>
> > The algorithm relies heavily on statistical independence tests. Therefore, the authors should provide more comprehensive robustness analysis and offer clear guidance on how to select the appropriate tests or parameters under different conditions.
>
> The choice of the statistical test is strongly constrained by the nature and dimensions of the data matrix. Keep in mind that the number of samples significantly decreases after each iteration of the ICSCM algorithm, since samples that are definitely classified by a previously selected rule do not count in the next statistical tests. This limits the applicability of statistical tests, as illustrated by the permutation test example below.
> The permutation test is a common approach for estimating the probability of observing a given distribution of values in vectors $e$ and $y$ under the null hypothesis H0. This test consists of:
> Compute the statistic associated with the observed distribution;
> Generate n (for example, n=10 000) other “observations” by permuting the values of one of the observed vectors (let say $y$);
> Compute the n values of the statistic associated with the n generated distributions;
> Estimate a p-value of how probable it is to have the statistic of the original observed distribution, given the n simulated values.
> In the case of a small sample size, the n iterations inevitably include some cases in which the permutation results in $y$ having a unique value. In these cases, the test statistic cannot be computed. Moreover, ignoring these cases would alter the distribution that the simulation intends to estimate. Therefore, the permutation tests do not solve the problem of computing p-values in very small sample sizes.
> No ideal solution exists for conducting this independence test under small-sample conditions. The chi-squared test has limitations in this regime, but it has the practical advantage of being consistently applicable; this consideration motivated our choice of this procedure.
>
> Regarding the selection of test hyperparameters, we refer the reader to our comment to reviewer D6h4 about the $\alpha$ parameter.

---

> > ### Author Response · Authors · 2026-06-17
> >
> > (second part of the previous comment)
> >
> > ---
> >
> > > I think for the method to be convincing to me (particularly regarding real-world relevance) I'd need to see that the method performs well against a range of more established methods in (primarily) causal discovery and secondly causal inference. I'm not expecting the method to outperform alternatives (because one naturally trades off biasedness against explainability) but nonethless it'd be good to understand the ball-park we're in. I understand that causal discovery methods per se have a wider task focus (e.g. whole graph, not just parents etc.) but as a practitioner I might be deciding between e.g. a combination of FCI / PC w/ non-parametric CI tests and/or Random forest vs ICSCM on its own, so it'd be really helpful if the authors could situation their method relative to an alternative pipeline that achieves an equivalent aim.
> >
> > As mentioned by the reviewer, the ISCM is not expected to outperform other methods in its causal retrieval abilities. Its main advantage is to apply to large data causal discovery settings. Following the reviewer’s suggestion, we intended to compare ICSCM to the Peter-Clark (PC) algorithm on the benchmark.
> >
> > The PC algorithm estimates the causal graph from observational data starting from a complete graph over all variables. It iteratively removes edges by testing conditional independences: an edge between variables $X_i$ and $X_j$ is removed if a conditioning set $S$ is found such that S creates conditional independence between $X_i$ and $X_j$. The remaining edges are then partially oriented into a Completed Partially Directed Acyclic Graph (CPDAG) using v-structure detection and Meek's orientation rules. Our implementation uses the results of this procedure to define a set of potential causal parents of the variable of interest $Y$. Then a Random Forest model is built on the training data restricted to this set. The models are named ‘PCRF’ in the results Figure presented here : https://anonymous.4open.science/r/icscm-expe-real-data-4328/figures/comparing-icscm-with-pc.pdf.
> >
> > The dimensions of the 6 datasets of the \texttt{tableshift} benchmark are presented in Table 1. Two factors strongly influence the computing time of PC: the number of variables (denoted $p$) and the maximum size of conditioning sets (denoted $s$). The computing time increases exponentially with both parameters, which limits our experiments to the 3 smaller datasets, with respectively $p=26$, $p=118$ and $p=183$ variables, with $s$ capped at $s=3$. Running PC with larger values of $p$ or $s$ would have required weeks-long executions. Additionally, the independence test used in our implementation of PC is a partial correlation test (that works under the assumption that the relation between variables is linear), as a non-parametric test would have strongly increased the computing time. This illustrates the scalability limitations of standard causal discovery methods such as PC in high-dimensional settings. ICSCM trades a stronger structural assumption than PC for better scalability. These two methods are therefore complementary and should be selected based on practical constraints.
> >
> > The results presented in Figure above show the causal retrieval scores of 4 repetitions of PC alongside the scores of SSCM and ICSCM from Figure 6. The PCRF algorithm first identifies a set of causal parents of $Y$ with PC, then trains a Random Forest on this set. In the experiments presented here, PC does not eliminate enough edges to remove variables from the potential causal parents of $Y$. Therefore, the Random Forest is trained on the whole dataset and does not benefit from the causal knowledge provided by PC. We consider that these experiments emphasize the practical limitations of algorithms like PC, which ICSCM circumvents by making a structural hypothesis and specializing in discovering categories of variables rather than the full graph.

---

### Review · Reviewer_txy4 · 2026-04-01

**Summary Of Contributions:**

The paper proposes a causal variation of set covering machines that incoporate Conditional Independence/CI tests to avoid selecting spurious/non-causal predictors.
As well as a theoretical result showing that the method can recover the causal parents of a target variable (which I haven't checked in detail but which I believe follows naturally from the theory concerning causal invariance), a modification of the Set Covering Machine (as mentioned above), and some simulation and empirical performance results.

**Audience:**

Yes

**Audience Explanation:**

Yes, SCMs are interesting and underexplored in their own right, and notwithstanding my comments below,  the causal angle makes for a compelling combination of interpretable decision tree-style functions with causal discovery.

**Claims And Evidence:**

Yes

**Claims Explanation:**

Broadly yes, although as per my remarks below I think the comparisons are very limited - e.g. the real-world eval compares ICSCM to SCM instead of broader causal discovery algos.

Secondly, although this I don't think is a great problem as it applies to much of the causal inference and causal discovery litearture, the causal features in the datasets are based on annotations rather than GT.

**Requested Changes:**

So firstly, to contextualise my review, I should briefly disclose my background. I have several years of experience both developing and applying causal inference and causal discovery methods across a range of domains. My strengths are more methodological and applied than deeply theoretical (so i can say upfront that my review of the theory here is light at best), and i'm slightly biased towards work that demonstrates credible real-world utility (e.g. via bencharmks /evals).

Set Covering Machines (SCMs) (as opposed to Structural Causal Models with which I am more familiar), have largely flown under my radar. Its kinda unfortunate because SCMs appear to occupy an interesting niche within interpretable machine learning, offering sparse, human-readable rule-based models.  As a consequence I'd say that I read this paper with considerable interest and also optimism.

So as mentioned, SCMs seem to have the potential to fulfil a unique role in interpretable and explainable machine learning, which is one of my primary areas of focus. That said, my prior is that non-ensemble methods in tabular settings typically significantly underperform stronger baselines such as random forests. My expectation is therefore that SCMs, by virtue of their restricted functional form (short conjunctions/disjunctions of rules), are unlikely to match the predictive performance of such models in general settings.

This limitation also raises questions about their downstream utility for causal tasks, where we are often trading off bias and sample efficiency. Do the authors agree with me so far?

Assuming I'm reading this correctly, my main high-level question is this: what gap is the proposed method intended to fill?

On one hand, the use of conditional independence tests suggests a form of causal discovery. However, this discovery process is tightly constrained by the (limited) hypothesis class defined by SCMs, meaning that only a restricted subset of conditional independences can be meaningfully represented and tested within the model.

On the other hand, the resulting SCM can be interpreted as a form of functional estimator that is constrained by these conditional independence checks and which could arguably then be used for causal inference (e.g. as a plugin estimator).

In this sense, the method appears to blur the line between causal discovery and causal inference/estimation (which is fine), but does so within a highly restricted model class. i.e. I wouldnt chose SCMs for causal inference because of the likely misspecification that would result.

If i take this logic further, it raises a concern about practical utility: if the true data-generating process deviates from the conjunctive/disjunctive structure assumed by SCMs, then BOTH the discovery component (via CI tests) AND the estimation component (via the learned rule set) are likely to be misspecified.

I realise that most causal discovery and most causal inference methods result in misspeification anyway, so I'm not saying that I'm expecting the perfect method, but I'm nonetheless a bit wary of SCMs as a model family overall because there are other approaches which should be much stronger by default.

I think for the method to be convincing to me (particularly regarding real-world relevance) I'd need to see that the method performs well against a range of more established methods in (primarily) causal discovery and secondly causal inference. I'm not expecting the method to outperform alternatives (because one naturally trades off biasedness against explainability) but nonethless it'd be good to understand the ball-park we're in.

I understand that causal discovery methods per se have a wider task focus (e.g. whole graph, not just parents etc.) but as a practitioner I might be deciding between e.g. a combination of FCI / PC w/ non-parametric CI tests and/or Random forest vs ICSCM on its own, so it'd be really helpful if the authors could situation their method relative to an alternative pipeline that achieves an equivalent aim.

---

> ### Author Response · Authors · 2026-06-17
>
> We thank the reviewer for the attentive reading and valuable comments on our work.
>
> Following the reviews, we uploaded a revised version of the manuscript. Here is a version of it where the changes are colored in red : https://anonymous.4open.science/r/icscm-expe-real-data-4328/ICSCM___TMLR___changes_after_reviews.pdf
>
> ---
>
> > Regarding the conjunctive functional form assumption (Assumption 2.4), I appreciate that the authors acknowledge its role as both a limitation and an enabler of the method's efficiency. That said, I am curious about the potential for generalization. Real-world causal mechanisms are often continuous or nonlinear, and it would be interesting to understand whether the core invariance-based criteria of Theorem 3.1 could be extended beyond the conjunction/disjunction family. For instance, could similar construction criteria be formulated in terms of more general graphical conditions, such as properties of the Markov blanket of $Y$? I recognize this may come at a cost to computational tractability; a brief discussion of the authors' intuitions on this direction would enrich the paper.
>
> > My expectation is therefore that SCMs, by virtue of their restricted functional form (short conjunctions/disjunctions of rules), are unlikely to match the predictive performance of such models in general settings. This limitation also raises questions about their downstream utility for causal tasks, where we are often trading off bias and sample efficiency. Do the authors agree with me so far?
>
> As the reviewer mentioned, the strong assumptions about the functional form limit the method's scope. That being said, Genomics-and Omics data in general- offers contexts where rule-based models successfully extract the information on the link between the observed variables ($x$) and the variable of interest ($y$). [1]
> In other domains, the suitability of SCMs is also context-dependent. Nevertheless, implementations of both the SCM and ICSCM algorithms are publicly available and can be evaluated without any domain-specific prior knowledge. Whether the model performs adequately relative to competing approaches will become apparent rapidly through standard predictive evaluation. Should the SCM prove to be a strong predictive model in a given setting, the application of ICSCM is then well-motivated and methodologically sound.
> It is worth noting that several studies have reported competitive performance of SCMs on real-world datasets [2]. This is consistent with the fact that SCMs constitute a special case of decision tree models, a family of methods whose appropriateness across a broad range of applied settings is well established in the machine learning literature. In the absence of prior knowledge of an SCM's suitability for a given dataset, a natural and practical approach is to fit the model and assess its adequacy using predictive performance metrics, with standard model comparison procedures as a diagnostic tool.
> [1] Godon, T., Plante, P., Corbeil, J., Germain, P., & Drouin, A. (2025). On Selecting Robust Approaches for Learning Predictive Biomarkers in Metabolomics Data Sets. Analytical chemistry.
> [2] Marchand, Mario and John Shawe-Taylor. “The Set Covering Machine.” J. Mach. Learn. Res. 3 (2003): 723-746.
>
> ---
>
> > My main high-level question is this: what gap is the proposed method intended to fill?
>
> There exist interpretable machine-learning methods that can be applied to retrieve informative features in large datasets. But they rely solely on the amount of predictive information, and are therefore sensitive to spurious correlations.
>
> There exist causally-informed methods that rely on invariance to avoid spurious associations. But the number of operations grows exponentially with the number of variables; therefore, they are inapplicable for large datasets such as Omics data.
>
> The ICSCM is a causally-informed algorithm that scales to large datasets.
>
> ---
>
> > In this sense, the method appears to blur the line between causal discovery and causal inference/estimation (which is fine), but does so within a highly restricted model class. i.e. I wouldnt chose SCMs for causal inference because of the likely misspecification that would result.
>
> Yes, we think it is better suited to causal discovery, since the ICSCM explores all possible decision rules based on the observed variables.
>
> ---
>
> > If i take this logic further, it raises a concern about practical utility: if the true data-generating process deviates from the conjunctive/disjunctive structure assumed by SCMs, then BOTH the discovery component (via CI tests) AND the estimation component (via the learned rule set) are likely to be misspecified.
>
> Yes, this is true. The ICSCM is expected to be useful when the data-generating process can be modelled by a conjunction/disjunction, and will be misspecified otherwise.
>
> ---
>
> (second part of rebuttal in another comment)

---

> > ### Author Response · Authors · 2026-06-26
> >
> > (second part of the previous comment)
> > [this response, sent on June 17th, was misplaced in the comments section of another review]
> >
> > ---
> >
> > > I think for the method to be convincing to me (particularly regarding real-world relevance) I'd need to see that the method performs well against a range of more established methods in (primarily) causal discovery and secondly causal inference. I'm not expecting the method to outperform alternatives (because one naturally trades off biasedness against explainability) but nonethless it'd be good to understand the ball-park we're in. I understand that causal discovery methods per se have a wider task focus (e.g. whole graph, not just parents etc.) but as a practitioner I might be deciding between e.g. a combination of FCI / PC w/ non-parametric CI tests and/or Random forest vs ICSCM on its own, so it'd be really helpful if the authors could situation their method relative to an alternative pipeline that achieves an equivalent aim.
> >
> > As mentioned by the reviewer, the ISCM is not expected to outperform other methods in its causal retrieval abilities. Its main advantage is to apply to large data causal discovery settings. Following the reviewer’s suggestion, we intended to compare ICSCM to the Peter-Clark (PC) algorithm on the benchmark.
> >
> > The PC algorithm estimates the causal graph from observational data starting from a complete graph over all variables. It iteratively removes edges by testing conditional independences: an edge between variables $X_i$ and $X_j$ is removed if a conditioning set $S$ is found such that S creates conditional independence between $X_i$ and $X_j$. The remaining edges are then partially oriented into a Completed Partially Directed Acyclic Graph (CPDAG) using v-structure detection and Meek's orientation rules. Our implementation uses the results of this procedure to define a set of potential causal parents of the variable of interest $Y$. Then a Random Forest model is built on the training data restricted to this set. The models are named ‘PCRF’ in the results Figure presented here : https://anonymous.4open.science/r/icscm-expe-real-data-4328/figures/comparing-icscm-with-pc.pdf.
> >
> > The dimensions of the 6 datasets of the \texttt{tableshift} benchmark are presented in Table 1. Two factors strongly influence the computing time of PC: the number of variables (denoted $p$) and the maximum size of conditioning sets (denoted $s$). The computing time increases exponentially with both parameters, which limits our experiments to the 3 smaller datasets, with respectively $p=26$, $p=118$ and $p=183$ variables, with $s$ capped at $s=3$. Running PC with larger values of $p$ or $s$ would have required weeks-long executions. Additionally, the independence test used in our implementation of PC is a partial correlation test (that works under the assumption that the relation between variables is linear), as a non-parametric test would have strongly increased the computing time. This illustrates the scalability limitations of standard causal discovery methods such as PC in high-dimensional settings. ICSCM trades a stronger structural assumption than PC for better scalability. These two methods are therefore complementary and should be selected based on practical constraints.
> >
> > The results presented in Figure above show the causal retrieval scores of 4 repetitions of PC alongside the scores of SSCM and ICSCM from Figure 6. The PCRF algorithm first identifies a set of causal parents of $Y$ with PC, then trains a Random Forest on this set. In the experiments presented here, PC does not eliminate enough edges to remove variables from the potential causal parents of $Y$. Therefore, the Random Forest is trained on the whole dataset and does not benefit from the causal knowledge provided by PC. We consider that these experiments emphasize the practical limitations of algorithms like PC, which ICSCM circumvents by making a structural hypothesis and specializing in discovering categories of variables rather than the full graph.

---

### Decision · Action_Editor_j1k4 · 2026-07-06

**Recommendation:** Accept with minor revision

**Audience:**

Yes

**Audience Explanation:**

The individuals on interpretable machine learning, causal discovery, and rule-based models would be interested in this paper.

**Claims And Evidence:**

Yes

**Claims Explanation:**

The reviewers agree that the paper makes a technically sound contribution. The theoretical claims are clearly stated, the proofs and experiments support the main conclusions, and the released code appears well organized and reproducible.

One reviewer raises concerns about the framing of the contribution. It is suggested to fix these minor issues in the camera-ready version.

---

> ### Author Response · Authors · 2026-07-22
>
> According to the decision, we uploaded a camera-ready version of the paper. It contains the improvements that followed the reviews, including Cautionary Note suggested to better frame the contribution, as well as the camera-ready TMLR formatting.
>
> We thank again the editors and reviewers for the time and attention given to our work.